# Competition between New Energy and Fuel Vehicles with Behavior-Based Pricing Strategies When Considering Environmental Concerns and Green Innovation

**Shaohua Chen** * and **Guomin Li**

College of Economics and Management, Taiyuan University of Technology, Taiyuan 030024, China; liguomin@tyut.edu.cn
* Correspondence: chenshaohua@tyut.edu.cn

**Abstract:** Environmental sustainability is an important issue in supply chain management (SCM). New energy vehicles (NEVs) have significant environmental value when compared to traditional fuel vehicles (FVs). Currently, there is intense competition between fuel and new energy vehicles, owing to differentiated pricing strategies. This paper focuses on behavior-based pricing (BBP) strategies between energy vehicles and fuel vehicles in a two-echelon supply chain wherein consumers are environmentally conscious. A two-period game-theoretic model is built to examine the effect of consumers' environmental concerns on competition between fuel and energy vehicles, behavior-based pricing strategies, supply chain efficiency, and social welfare. The analytical results indicate that consumers' environmental concerns facilitate the market share competition by new energy vehicle firms in the second period. If consumers care more about the environment, supply chain efficiency is improved in cases of retail as well as wholesale-and-retail behavior-based pricing strategies. Wholesale-and-retail behavior-based pricing strategies benefit all members of the supply chain, but this is not the case for retail behavior-based pricing strategies. If consumers are sufficiently concerned about the environment and new energy vehicle firms are more efficient, a win–win–win scenario for firms, consumers, and social welfare occurs in the two behavior-based pricing strategies. Counterintuitively, green innovation improves new energy vehicle, fuel vehicle as well as overall supply chain efficiency, in three cases.

**Keywords:** sustainable supply chain management; behavior-based pricing; new energy vehicles; environmental concerns; green innovation; duopoly competition

## 1. Introduction

The automotive industry is undergoing a technological revolution in new energy vehicles (NEVs) and energy transition to meet sustainability targets. The most obvious advantage of new energy vehicles over traditional fuel vehicles (FVs) is that they are environmentally friendly [1]. This reduces the dependence of automobiles on fossil fuels, which is conducive to alleviating the oil crisis and eventually transforming the energy structure of society [2]. Further, NEVs offer the added advantages of low noise levels and greater comfort. Therefore, energy vehicles will play a vital role in the future development of the automobile industry.

However, FVs still occupy a large share of the current market, creating significant competition for NEVs. JATO Dynamics, a specialized consultancy for the global automotive industry, released its best-selling single-model report for Q1 2023. This shows that the traditional FV market is still dominant, though the demand for NEVs is also rising. One competitive strategy in the automotive industry is behavior-based pricing (BBP), which is commonly used in vehicle industries [3]. As per the BBP strategy, new customers are charged lower prices than previous customers with the help of consumer purchase history

data. For instance, Audi offers incentives to its dealers to sell new cars to customers owning automobiles of other brands, such as Mercedes-Benz, Infiniti, Lexus or BMW [4].

Nowadays, people are becoming increasingly environmentally conscious, and therefore preferring to purchase new energy vehicles. When people previously owning fuel vehicles switch to new energy vehicles, they gain perceived utility from environmental concerns. Thus, when firms dealing in fuel vehicles and new energy vehicles use BBP strategies to compete for market share, consumers experience increases and decreases in utility due to environmental concerns, respectively. Therefore, this paper explores whether consumers' environmental concerns affect firms' BBP strategies, profits, supply chain efficiency, and social welfare.

Green innovation can effectively reduce the negative impact of environmental pollution, thereby promoting sustainable development. Unlike traditional innovation, which focuses only on economic benefits, green innovation can benefit both consumers and firms and significantly reduce negative impacts on the environment. It helps to balance economic and environmental benefits. Green innovation by NEV firms creates environmental value. Therefore, to a certain extent, the environmental value of NEVs reflects NEV firms' level of green innovation. This paper also examines the impact of green innovation on NEV and overall supply chain efficiency.

Therefore, the paper answers the following research questions (RQs):

**RQ 1:** *How do consumers' environmental concerns affect the competition between NEV and FV firms and their BBP strategies?*

**RQ 2:** *How do consumers' environmental concerns and firms' green innovation affect overall supply chain efficiency?*

**RQ 3:** *How do consumers' environmental concerns affect their surplus and social welfare?*

In addition, this paper examines another research question using an empirical approach.

**RQ 4:** *How do consumers' environmental concerns affect the economy?*

A two-period game-theoretic model wherein two manufacturers sell new energy vehicles and fuel vehicles to consumers through independent retailers in a two-echelon supply chain is considered. In the first period, new energy vehicle and fuel vehicle firms obtain consumer purchase data through sales, which, in turn, leads to differentiated pricing for new and existing customers in the second period to compete for market share. As consumers are environmentally conscious, shifting between FVs and NEVs generates an increase or decrease in utility based on environmental concerns. The study yields several important and interesting findings. Wholesale-and-retail BBP is always more beneficial than retail BBP for supply chain members. If consumers focus more on the environment, supply chain efficiency improves for both types of BBP. Higher environmental concerns of consumers and more efficient new energy vehicle firms result in a win–win–win scenario for firms, consumers, and social welfare for both types of BBP. Green innovation by new energy vehicle firms benefits supply chain efficiencies for both NEV and FV.

Nowadays, new energy vehicles are an important trend in the development of the automobile industry and an important issue of academic concern. The main purpose of the paper is to explore the impacts of consumers' environmental concerns and green innovations of new energy vehicle manufacturers on the competition between NEV and FV firms, and then to explore the impacts on supply chain efficiency and social welfare.

The contributions of this study mainly pertain to two aspects. First, this paper examines BBP strategies in a two-echelon supply chain consisting of fuel vehicle and new energy vehicle manufacturers and retailers. The analytical model verifies that consumers' environmental concerns and green innovation significantly influence supply chain efficiency, which has not been considered in existing research. In addition, the specific effects of environmental concerns on duopoly competition, BBP strategies, and social welfare through model analysis are obtained, providing reasonable suggestions for firms.

The remainder of this paper is organized as follows. Section 2 reviews the related literature. Section 3 presents a model of interactions between consumers and firms. Section 4 analyzes the strategies for two types of BBP (retail and wholesale-and-retail) based on consumers' environmental concerns. Section 5 presents the results and analysis of the basic model. In Section 6, the paper extends the basic model for two aspects. Section 7 provides the empirical analysis and validation. Finally, Section 8 concludes the paper.

## 2. Literature Review

First, this study is closely related to the pricing of new energy vehicles. New energy vehicles provide an important direction for the future development of the automotive industry [5–8]. Li et al. (2020) [1] investigated the impact of subsidies and dual credit policies on the production decisions for new energy and fuel vehicles, considering battery recycling in a competitive environment. Subsequently, Jiao et al. (2022) [2] showed that subsidies, dual credit policies, and charging pile construction positively affect new energy vehicle diffusion, and controlling the trading price of new energy credit within a certain range, as proposed in this study, is vital for maximizing policy effectiveness. Zhao et al. (2022) [9] constructed supply chain models for new energy and traditional vehicles, compared the pricing, demand, and supply chain profit of the two types of vehicles under decentralized and centralized decision-making, and designed a revenue-sharing and ex-factory price negotiation contract mechanism to coordinate the supply chain under certain conditions. Liu et al. (2023) [10] considered a monopoly automaker under dual-credit and subsidy backslope policies to explore the automaker's optimal product-line strategies by constructing a stylized pricing model that considers heterogeneous consumption preferences. In addition, Heydari et al. (2021) [11] contribute to the literature by providing an analytical approach to examine the channel coordination and pricing issues in a green supply chain under consumers' environmental awareness when considering the product's green quality. Liao et al. (2022) [12] examine competition between new energy vehicles and traditional fuel, and analyze the role of governmental policy on competition and the impacts of different regulation intensities on the promotion effect of NEVs. Subsequently, Lera-Romero et al. (2024) [13] introduce a general version of the time-dependent electric vehicle routing problem with time windows for electric vehicles (EVs), which incorporates the time-dependent nature of the transportation network both in terms of travel times and energy consumption. This paper investigates BBP strategies for NEV and FV supply chains and examines the effects of consumers' environmental concerns on supply chain efficiency.

Second, the relevant literature focuses on behavior-based pricing. The literature has focused on the impact of poaching behavior on firms' profits [14,15], pricing [16,17], and social welfare. This stream of research has shown that firms with consumer recognition obtain lower profits in a monopoly with overlapping generations of forward-looking customers [18]; the poaching price is lower than the repeat price, and patient consumers intensify competition, while patient firms soften competition in a duopoly [19]; and poaching leads to switching and inefficient social welfare in a duopoly [20]. Rhee and Thomadsen (2016) [21] studied behavior-based pricing in a vertical duopoly and showed that firms at a competitive disadvantage obtain higher profits with BBP, when the discount factor is small.

Studies also demonstrate how behavior-based price discrimination affects quality [3,22]. Jing (2016) [23] examined the impact of behavior-based price discrimination on firms' endogenous quality differentiation and profits in a vertical duopoly. Other studies related to two-period dynamic pricing include Zhang et al. (2020) [24] and Chen & Jiang (2021) [25]. In contrast, this study focuses on the effect of consumers' environmental concerns on the behavior-based pricing of vertically differentiated products in a two-echelon supply chain. This study contributes to this stream of literature by analyzing how firms' competition and profitability are affected when consumers have environmental concerns.

Third, the relevant body of literature focuses on firms' green innovation. James was the first to define green innovation, indicating that the basic goal of green innovation is to reduce negative environmental impacts; at the same time, it is a new process or

product that contributes to enhancing a company's own value [26]. Green innovation has the following six main characteristics: the object of innovation, market positioning, environmental impact, cycle stage, innovation intent, and level of innovation [27]. It has also been argued that green innovation encompasses not only innovative activity related to green products and processes, but also technological innovations for energy conservation, pollution prevention, product design, and environmental management [28]; green innovation denotes innovation which is environmentally friendly and contributes to the sustainability of environmental resources [29]; green innovation includes a combination of factors related to ecological, technological, market demand, regulatory compliance, and suppliers that drive a company's ability to explore and apply innovation [30]. In addition, research on green innovation includes that on relationships between its various practices as well as consumer resistance to it [31], on green innovation as a significant positive predictor of corporate sustainable development [32], the relationship between digitalization and green innovation [33], green technology innovation in manufacturing under the influence of environmental regulation [34], and the mediating role of corporate green innovation in the role of fintech innovation and green transformational leadership in improving corporate environmental performance [35]. This study considers the impact of green innovation in new energy vehicles and public environmental concerns on firms' pricing strategies and market competition.

Finally, the work contributes to the literature on duopoly competition. Li et al. (2019) [36] investigated offensive pricing strategies in a duopoly supply chain, wherein two platforms offer highly dependent products and identify the impact of message dissemination on offensive pricing strategies. Jiang & Liu (2019) [37] found that managerial optimism about demand for one duopolistic firm can increase both firms' profits. In a two-echelon duopoly supply chain, Jin et al. (2019) [38] identified whether to integrate with a supplier and how much to invest in reducing manufacturer cost in a duopoly. Additionally, the related literature includes research by Chen & Jia (2020) [39], Jia & Zhang (2013) [40], Ghosh & Shah (2015) [41], Kwark et al. (2017) [42], Wu (2019) [43], Sim et al. (2019) [44], Xu et al. (2020) [45], Xu et al. (2022) [46], and Babic et al. (2022) [47]. This paper studies how consumers' environmental concerns and green innovation affect supply chain efficiency in duopoly competition. To the best of our knowledge, this is the first study addressing this issue.

This paper examines the effect of consumers' environmental concerns and green innovation on competition between fuel and energy vehicles, behavior-based pricing strategies, supply chain efficiency, and social welfare. The novelty of this study lies in the research questions and the findings. Consumers' environmental concerns can improve the market competitiveness of new energy vehicles, as well as the green innovation level of new energy vehicle firms, which in turn improves their performance. In addition, a counterintuitive conclusion was obtained in that green innovation can improve not only the supply chain efficiency of new energy vehicles, but also that of fuel vehicles and the overall supply chain efficiency. This study can provide suggestions on market investigation and green innovation technology development for new energy vehicle firms in terms of practice, in order to help new energy vehicle firms gain greater market competitiveness and higher supply chain efficiency through the BBP strategies.

## 3. Model Setup

This study firstly analyzes the competition between new energy vehicle and fuel vehicle firms using analytical and theoretical models, and obtains the theoretical results of consumers' environmental concerns on the competition. This empirical analysis can validate the conclusion of the analytical model by analyzing the actual data to enhance the practical value of the study. The exploration of the relationship between data and variables helps us to reveal the development advantages brought by consumers' environmental concerns for NEV firms and the underlying mechanism.

In the empirical part, regression models controlling individual effects and fixed effects were used for testing. There are some unobserved individual features and time trends in the panel data, which may be related to explanatory variables or explained variables, thus affecting the model's estimation results. By introducing individual and year-specific intercept terms, these unobservable heterogeneity and time trends can be controlled. Therefore, using this model can minimize the estimation bias and obtain more reliable and effective empirical analysis results.

The paper begins by describing manufacturers and retailers of new energy vehicles and fuel vehicles, consumer utility, and the timing of the game as follows.

### 3.1. Manufacturers and Retailers

A two-period competitive duopoly in a two-echelon supply chain is considered. Two manufacturers, $M_L$ and $M_H$, offer fuel vehicles (FV) and new energy vehicles (NEV) through their respective retailers, $R_L$ and $R_H$, to consumers in each period, as shown in Figure 1. Assume that new energy vehicles and fuel vehicles have the same functional performance. However, the quality of new energy vehicles is higher because they acquire environmental or green value from green innovation. We denote the quality of new energy vehicles and fuel vehicles as $s_H$ and as $s_L$, respectively, where $s_H > s_L$. The difference in quality, defined as $s_\Delta \equiv s_H - s_L > 0$, indicates the environmental value of new energy vehicles and, to a certain extent, reflects the level of *green innovation*. Vehicle differentiation can also be measured by using marginal costs. Assume the marginal costs of a new energy vehicle and fuel vehicle to be $c_H$ and $c_L$, respectively. Without loss of generality, the marginal cost of the fuel vehicle is normalized to zero. Thus, the difference between the marginal costs of new energy vehicles and fuel vehicles is denoted as $c_\Delta \equiv c_H - c_L = c_H$.

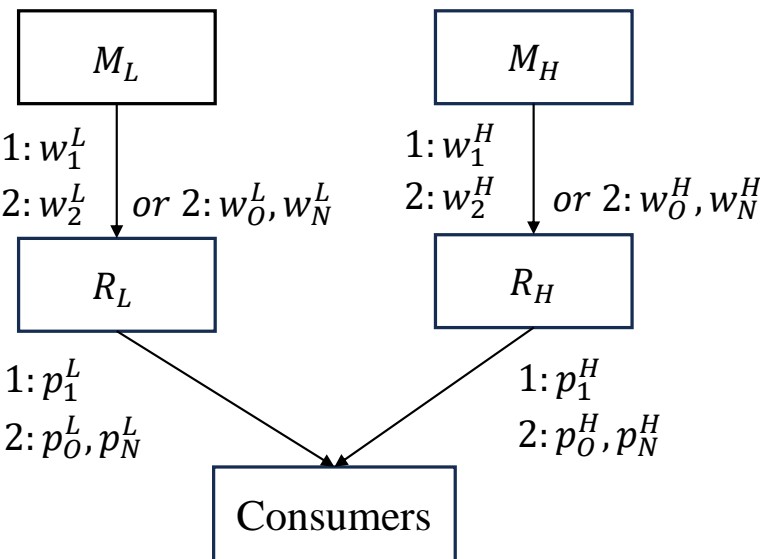

**Figure 1.** The supply chain structure.

Let $\mu \equiv \frac{c_H - c_L}{s_H - s_L} = \frac{c_\Delta}{s_\Delta}$. As Jing (2016) [23] noted, the ratio represents the firms' relative production efficiency: as the ratio increases, the relative production efficiency of FV firms increases and that of NEV firms decreases. Rhee and Thomadsen (2016) [21] assumed that $\mu \in [0, 1]$, and Jing (2016) [23] considered $\mu > 0$. In fact, if FV firms are more efficient than NEV firms, $\mu$ may be greater than one. Therefore, assume $\mu > 0$ for more general scenarios and $\mu \in [0, \overline{\mu}]$. This ratio will occur in many results below. It is called quality-adjusted cost.

### 3.2. Consumers

Without a loss of generality, a unit mass of consumers is available in the market. Each consumer purchases, at most, one vehicle during each period. The base value of the

vehicle, denoted as $V$, is sufficiently high for the market to be fully covered. Consumer preferences for vehicle quality are denoted by $\theta$ ($\theta \in [0, \overline{\theta}]$). When a consumer values quality, his/her preference $\theta$ is high; when he/she does not care about quality, his/her preference is low. The cumulative distribution function of consumer tastes is denoted as $F(\theta)$, and the probability density function is denoted as $f(\theta)$. The inverse hazard rate is $H(\theta) = \frac{1-F(\theta)}{f(\theta)}$. Without loss of generality, the paper focuses on the situation wherein $\theta$ is a uniform distribution on $[0, 1]$.

The surplus when a customer consumes one unit of Vehicle $i$ is $U(\theta) = V + \theta s_i - p_t^i$, where $i \in \{L, H\}$, $t \in \{1, 2\}$. Vehicle $L$ denotes an FV, and Vehicle $H$ denotes an NEV. Following [14,23], a consumer of type-$\theta$ is willing to pay up to $\theta s_i$ for a unit of Vehicle $i$, that is, the consumer's type is defined by his marginal willingness to pay for the incremental quality of the vehicle. $p_t^i$ is the retail price of Vehicle $i$ in period $t$. Furthermore, $p_O^i$ and $p_N^i$ represent the retail prices of Vehicle $i$ for old and new consumers, respectively.

The consumers' degree of environmental concern was measured using $\lambda$ ($\lambda \in [0, 1]$). Assume that the $\lambda$ is constant for all customers. For instance, when consumers purchase Vehicle $L$ in period 1 and switch to vehicle $H$ in period 2, they obtain a positive utility, or utility gain, from the environmental value. The magnitude of the utility gain is $\lambda s_\Delta$. Assume that the discount factor for consumers and firms is one in period 2.

### 3.3. Timing of the Game

The timing of the game is as shown in Figure 2: In period 1, there are three stages. In stage 1, manufacturers $L$ (or $M_L$) and $H$ (or $M_H$) simultaneously set wholesale prices $w_1^L$ and $w_1^H$ for the first period. In stage 2, retailers $L$ (or $R_L$) and $H$ (or $R_H$) simultaneously set the retail prices $p_1^L$ and $p_1^H$ for the first period. In stage 3, consumers decide which vehicle to choose after observing the retail prices. Period 2 comprised three stages. In stage 1, if manufacturers do not adopt BBP, they simultaneously set wholesale prices $w_2^L$ and $w_2^H$ for the second period. If manufacturers adopt BBP, they set wholesale prices discriminately for retailers selling to repeat and new customers. Denote the wholesale prices for retailers to sell to old buyers by $w_O^L$ and $w_O^H$ and those to new buyers by $w_N^L$ and $w_N^H$. In stage 2, retailers simultaneously set the repeat retail price, denoted by $p_O^L$ and $p_O^H$, for repeat customers and the poach retail price, denoted by $p_N^L$ and $p_N^H$, for switchers. In stage 3, the customers decide whether to consume the same vehicle or switch to a competing vehicle, after observing the repeat and poaching prices. Consumers maximize their total expected utility, while firms maximize their profits. Table 1 summarizes all the notations.

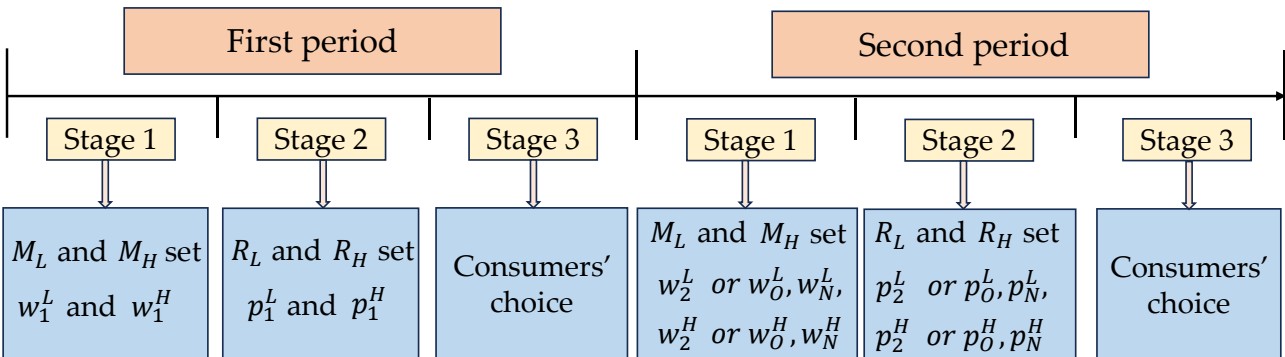

**Figure 2.** The timing of the game.

**Table 1.** Notations ($i \in \{L, H\}$, $t \in \{1, 2\}$).

| Notations | Description |
|---|---|
| $V$ | Consumers' base value of the vehicle |
| $\theta$ | Consumers' taste for the quality of the vehicle |
| $\lambda$ | Consumers' degree of environmental concerns |
| $s_i$ | The quality of the Vehicle $i$ |
| $c_i$ | The marginal cost of the Vehicle $i$ |
| $M_i$ | The manufacturer $i$ |
| $R_i$ | The retailer $i$ |
| $w_t^i$ | The wholesale price of the Vehicle $i$ in period $t$ |
| $w_O^i$ | The wholesale price of the Vehicle $i$ for old consumers in period 2 |
| $w_N^i$ | The wholesale price of the Vehicle $i$ for new consumers in period 2 |
| $p_t^i$ | The retail price of the Vehicle $i$ in period $t$ |
| $p_O^i$ | The retail price of the Vehicle $i$ for old consumers in period 2 |
| $p_N^i$ | The retail price of the Vehicle $i$ for new consumers in period 2 |
| $\Pi_M^i \left( \Pi_R^i \right)$ | Manufacturer (Retailer) $i$'s total profits over two periods |
| $\Pi_{Mt}^i \left( \Pi_{Rt}^i \right)$ | Manufacturer (Retailer) $i$'s profits in period $t$ |

## 4. BBP Strategies of NEV and FV Firms

First, the equilibrium without BBP as a benchmark case is analyzed. In this case, the two-period game is a repetition of the static game. Superscript 0 represents the benchmark case. $U_1^L \left( \theta_1^0 \right) = U_1^H \left( \theta_1^0 \right)$, where $\theta_1^0$ is the preference for quality of the marginal customer between purchasing vehicle $L$ at price $p_1^{L^0}$ and vehicle $H$ at price $p_1^{H^0}$. Hence,

$$\theta_1^0 s_L - p_1^{L^0} = \theta_1^0 s_H - p_1^{H^0} \tag{1}$$

$$\theta_1^0 \left( p_1^{H^0}, p_1^{L^0} \right) = \left( p_1^{H^0} - p_1^{L^0} \right) / s_\Delta \tag{2}$$

The retailers' payoff functions are

$$\Pi_{R1}^{L^0} \left( p_1^{L^0} \right) = \left( p_1^{L^0} - w_1^{L^0} \right) F \left( \theta_1^0 \right) \tag{3}$$

$$\Pi_{R1}^{H^0} \left( p_1^{H^0} \right) = \left( p_1^{H^0} - w_1^{H^0} \right) \left[ 1 - F \left( \theta_1^0 \right) \right] \tag{4}$$

The manufacturers' payoff functions are

$$\Pi_{M1}^{L^0} \left( w_1^{L^0} \right) = w_1^{L^0} F \left( \theta_1^0 \right) \tag{5}$$

$$\Pi_{M1}^{H^0} \left( w_1^{H^0} \right) = \left( w_1^{H^0} - c_H \right) \left[ 1 - F \left( \theta_1^0 \right) \right] \tag{6}$$

The equilibrium results for the benchmark case are summarized in Table A1 in Appendix A.

Following [4], this paper discusses two types of BBP: retail and wholesale-and-retail. If manufacturers do not have access to consumer purchase history data, they cannot set wholesale prices for retailers to sell to old or new customers. Only retailers use BBP to charge behavior-based retail prices for repeat and switch buyers. Refer to this case as *retail BBP*. If manufacturers can also use customer purchase history data, they price discriminately to set wholesale prices for retailers who sell to old and new customers. In other words, both, manufacturers and retailers adopt BBP, which is called *wholesale-and-retail BBP*.

### 4.1. Retail BBP with Environmental Concerns

Consumers' environmentally friendly utility gains or losses when purchasing different vehicles in two stages are considered. For instance, when a consumer purchases an FV in the first period and an NEV in the second period, he or she will have positive environmentally

friendly utility. This paper further analyzes the impact of consumers' environmental concerns on firms' adoption of BBP strategies.

When only retailers adopt BBP, this paper solves the subgame-perfect equilibrium of the two-period game backward and first analyzes period 2, taking the first-period market segmentation as given.

*Period* 2. Discuss the competition on *L*'s turf and *H*'s turf, respectively. Figure 3 shows the market segmentation for each period. A customer on *L*'s turf ($F(\theta) \in [0, F(\theta_1)]$) will repeat the purchase vehicle *L* if $U_2^L(\theta) \geq U_2^H(\theta)$, i.e.,

$$\theta s_L - p_O^L \geq \theta s_H + \lambda(s_H - s_L) - p_N^H \tag{7}$$

that is,

$$\theta \leq \left(p_N^H - p_O^L\right)/(s_H - s_L) - \lambda \tag{8}$$

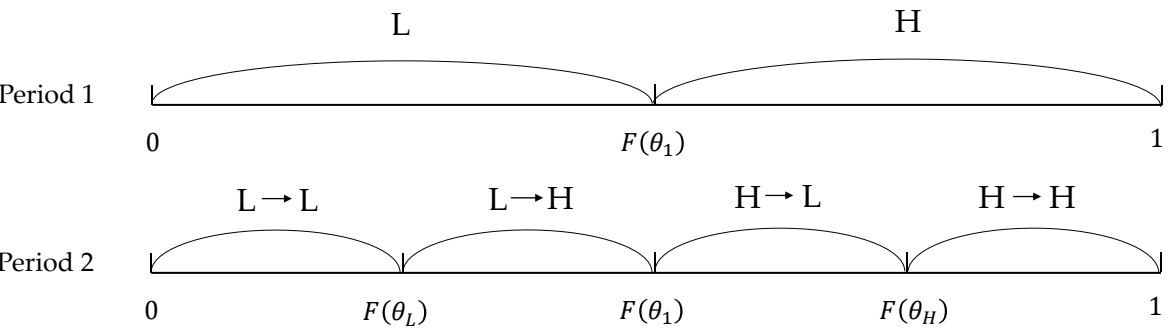

**Figure 3.** The market segmentation in each period.

Here, set $\theta_L = \left(p_N^H - p_O^L\right)/(s_H - s_L) - \lambda$, where $\theta_L$ denotes the preference for the quality of a marginal customer who is indifferent between repurchasing vehicle *L* and switching to vehicle *H*. The larger the value of $\theta_L$, the more consumers will repeat the purchase of FVs in the second period. The left-hand side of Equation (7) denotes the utility of a consumer who purchases a vehicle *L* in period 2 at repeat price, $p_O^L$, and the right-hand side denotes the utility of a consumer switching to vehicle *H* in period 2 at poaching price, $p_N^H$.

Likewise, a customer on *H*'s turf ($F(\theta) \in [F(\theta_1), 1]$) will repeat the purchase vehicle *H* if $U_2^H(\theta) \geq U_2^L(\theta)$, that is,

$$\theta \geq \left(p_O^H - p_N^L\right)/(s_H - s_L) - \lambda \tag{9}$$

Set $\theta_H = \left(p_O^H - p_N^L\right)/(s_H - s_L) - \lambda$, where $\theta_H$ denotes the quality preference of the marginal customer who is indifferent between purchasing vehicle *H* and switching to vehicle *L*. A greater $\theta_H$ indicates that fewer consumers purchase NEVs in the second period. When consumers purchase vehicle *H* in period 1 and switch to vehicle *L* in period 2, they experience an environmentally friendly utility loss. The magnitude of utility loss is $\lambda(s_H - s_L)$. The retailer's profit function in period 2 is as follows:

$$\Pi_{R2}^L\left(p_O^L, p_N^L\right) = \left(p_O^L - w_2^L\right)F(\theta_L) + \left(p_N^L - w_2^L\right)[F(\theta_H) - F(\theta_1^*)] \tag{10}$$

$$\Pi_{R2}^H\left(p_O^H, p_N^H\right) = \left(p_O^H - w_2^H\right)[1 - F(\theta_H)] + \left(p_N^H - w_2^H\right)[F(\theta_1^*) - F(\theta_L)] \tag{11}$$

where $w_2^L$ and $w_2^H$ are the uniform wholesale prices charged by manufacturers $M_L$ and $M_H$ in period 2, respectively. The manufacturer's profit function in period 2 is as follows:

$$\Pi_{M2}^L\left(w_2^L\right) = w_2^L[F(\theta_L) + F(\theta_H) - F(\theta_1^*)] \tag{12}$$

$$\Pi_{M2}^{L}\left(w_2^H\right) = \left(w_2^H - c_H\right)\left[F(\theta_1^*) - F(\theta_L) + 1 - F(\theta_H)\right] \tag{13}$$

Hence, the following results are obtained. The repeat retail prices are as follows:

$$p_O^{L*} = w_2^L + \frac{s_\Delta F(\theta_L)}{f(\theta_L)}, \tag{14}$$

$$p_O^{H*} = w_2^H + \frac{s_\Delta}{H(\theta_H)} \tag{15}$$

The poach retail prices are

$$p_N^{L*} = w_2^L + \frac{s_\Delta\left[F(\theta_H) - F(\theta_1^*)\right]}{f(\theta_H)}, \tag{16}$$

$$p_N^{H*} = w_2^H + s_\Delta\left[\frac{1}{H(\theta_L)} - \frac{1 - F(\theta_1^*)}{f(\theta_L)}\right] \tag{17}$$

The wholesale prices in period 2 are

$$w_2^{L*} = \frac{-F(\theta_L) - F(\theta_H) + F(\theta_1^*)}{K}, \tag{18}$$

$$w_2^{H*} = c_H + \frac{F(\theta_L) + F(\theta_H) - F(\theta_1^*) - 1}{K} \tag{19}$$

where $K = f(\theta_L)\left(\partial\theta_L/\partial w_2^{L*}\right) + f(\theta_H)\left(\partial\theta_H/\partial w_2^{L*}\right)$.

The marginal customers are at

$$\theta_L^* = \frac{1}{s_\Delta}\left(w_2^{H*} - w_2^{L*} + s_\Delta\Phi_1\right) - \lambda, \tag{20}$$

where $\Phi_1 = \frac{1}{H(\theta_L^*)} - \frac{1 - F(\theta_1^*) + F(\theta_L^*)}{f(\theta_L^*)}$,

$$\theta_H^* = \frac{1}{s_\Delta}\left(w_2^{H*} - w_2^{L*} + s_\Delta\Phi_2\right) - \lambda \tag{21}$$

where $\Phi_2 = \frac{1}{H(\theta_H^*)} - \frac{F(\theta_H^*) - F(\theta_1^*)}{f(\theta_H^*)}$.

The equilibrium results for retail BBP are summarized in Table A2 in Appendix A.

**Proposition 1.** *(Switching and profits in period 2.) As customers become more concerned about the environment (i.e., as $\lambda$ increases) in the case of retail BBP,*

(a) *The wholesale, repeat, and poach prices of NEV increase (i.e., $\partial w_2^{H*}/\partial\lambda > 0$, $\partial p_O^{H*}/\partial\lambda > 0$, and $\partial p_N^{H*}/\partial\lambda > 0$); the wholesale, repeat, and poach prices of FV decrease (i.e., $\partial w_2^{L*}/\partial\lambda < 0$, $\partial p_O^{L*}/\partial\lambda < 0$, and $\partial p_N^{L*}/\partial\lambda < 0$).*

(b) *Furthermore, more consumers switch to NEV from FV than the other way round.*

(c) *Second-period profits from $R_H$ and $M_H$ increase, whereas those from $R_L$ and $M_L$ decrease.*

The proof is provided in Appendix B. Intuition suggests that consumers' concern about the environment increases their willingness to switch to new energy vehicles and decreases their willingness to switch to fuel vehicles. Thus, the poaching power of new energy vehicle firms improves, while that of fuel vehicle firms declines. Consequently, as their concerns become stronger, more consumers switch to new energy vehicles from fuel vehicles and fewer consumers switch to fuel vehicles from new energy vehicles. Furthermore, the new energy vehicle retailer obtains higher second-period profits than the fuel vehicle retailer.

Hence, when consumers pay more attention to the environment, new energy vehicle firms obtain a larger market share than fuel vehicle firms. In other words, consumer concerns about the environment are conducive to new energy vehicle firms poaching the market during the second period. Figure 4 shows that the prices of new energy vehicles and fuel vehicles in period 2 vary with environmental concerns. Here, set $\mu = 3/10$ and $s_\Delta = 1/2$. They can be any value within their respective ranges.

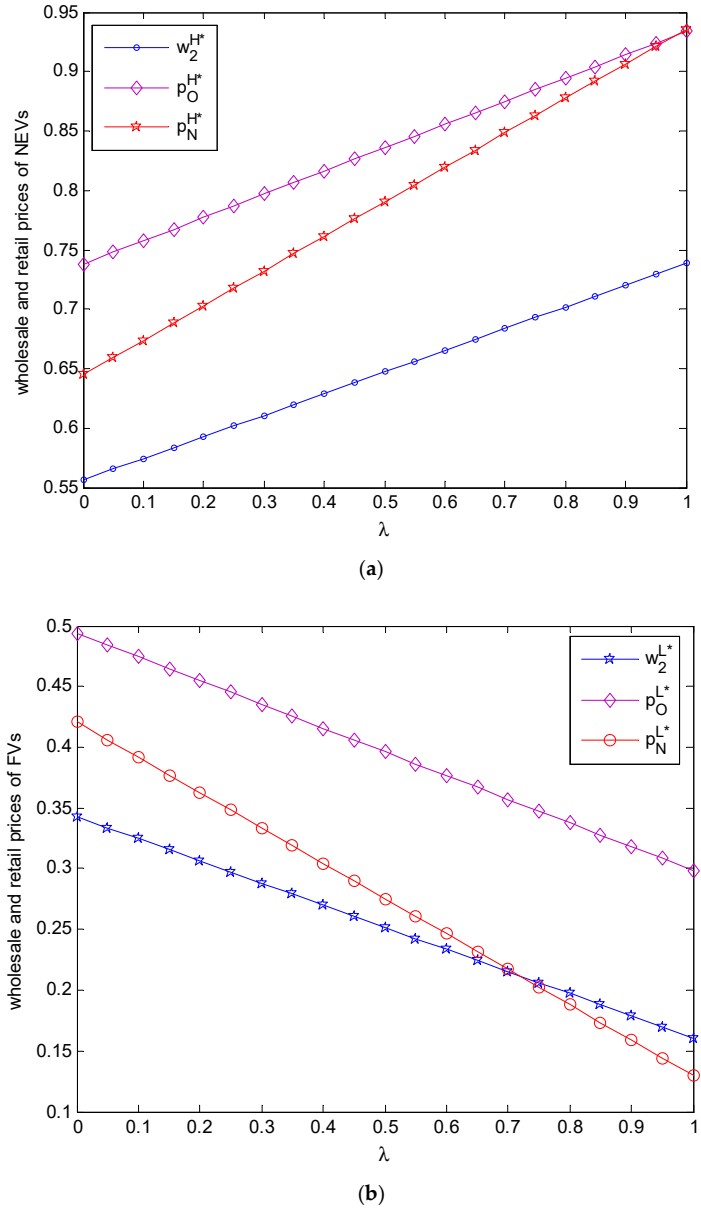

**Figure 4.** Prices vary with environmental concerns. (**a**) NEV prices; (**b**) FV prices.

***Period 1.*** Assume that consumers are strategic and have rational expectations of prices in period 2: That is, their expected prices are those obtained in period 2, which is consistent with the extant BBP literature [4,16,20,22]. Each consumer maximizes his/her total expected utility over the two periods. Identify the marginal consumer $\theta_1$, who is indifferent between (1) purchasing an FV in period 1 and switching to an NEV in period 2 and (2) purchasing an NEV in period 1 and switching to an FV in period 2. We obtain that

$$\theta_1 s_L - p_1^L + \left[\theta_1 s_H + \lambda(s_H - s_L) - p_N^{H*}\right] = \theta_1 s_H - p_1^H + \left[\theta_1 s_L - \lambda(s_H - s_L) - p_N^{L*}\right] \quad (22)$$

$\theta_1^*$ is determined by the following equation.

$$p_1^H - p_1^L + 2\lambda s_\Delta = p_N^{H*}(\theta_1^*) - p_N^{L*}(\theta_1^*) \tag{23}$$

Retailers and manufacturers maximize their total expected profit over the two periods. The retailers' payoff functions in period 1 are $\Pi_R^L(p_1^L) = (p_1^L - w_1^L)F(\theta_1^*) + \Pi_{R2}^{L*}$ and $\Pi_R^H(p_1^H) = (p_1^H - w_1^H)[1 - F(\theta_1^*)] + \Pi_{R2}^{H*}$. The manufacturer's payoff functions in period 1 are $\Pi_M^L(w_1^L) = w_1^L F(\theta_1^*) + \Pi_{M2}^{L*}$ and $\Pi_M^H(w_1^H) = (w_1^H - c_H)[1 - F(\theta_1^*)] + \Pi_{M2}^{H*}$.

*4.2. Wholesale-and-Retail BBP with Environmental Concerns*

Wholesale-and-retail BBP is observed in the automobile industry. For example, Audi and Chrysler offer incentives to their dealers to sell new cars to new customers to poach the market share of their competitors. When manufacturers also obtain consumer purchase history data, they can set different wholesale prices for retailers to sell to repeat and new buyers. Now analyze the case wherein both manufacturers and retailers adopt BBP. How do manufacturers and retailers adjust their BBP strategies while considering consumer concerns about the environment?

*Period 2.* The analysis in this period parallels the case of retail BBP. The marginal consumer on $L$'s turf is $\theta_L = (p_N^H - p_O^L)/(s_H - s_L) - \lambda$, and the marginal consumer on $H$'s turf is $\theta_H = (p_O^H - p_N^L)/(s_H - s_L) - \lambda$. The retailers' payoff functions in period 2 become

$$\Pi_{R2}^L(p_O^L, p_N^L) = (p_O^L - w_O^L)F(\theta_L) + (p_N^L - w_N^L)[F(\theta_H) - F(\theta_1^*)]. \tag{24}$$

$$\Pi_{R2}^H(p_O^H, p_N^H) = (p_O^H - w_O^H)[1 - F(\theta_H)] + (p_N^H - w_N^H)[F(\theta_1^*) - F(\theta_L)]. \tag{25}$$

where $w_O^L$ and $w_O^H$ are the wholesale prices charged by manufacturers for retailers selling to old buyers, and $w_N^L$ and $w_N^H$ are for selling to new buyers. The manufacturer's profit function in period 2 is as follows:

$$\Pi_{M2}^L(w_O^L, w_N^L) = w_O^L F(\theta_L) + w_N^L[F(\theta_H) - F(\theta_1^*)]. \tag{26}$$

$$\Pi_{M2}^H(w_O^H, w_N^H) = (w_O^H - c_H)[1 - F(\theta_H)] + (w_N^H - c_H)[F(\theta_1^*) - F(\theta_L)]. \tag{27}$$

The equilibrium results for the case of wholesale-and-retail BBP are summarized in Table A3 in Appendix A.

**Proposition 2.** *(Switching and profits in period 2.) As $\lambda$ increases in wholesale-and-retail BBP,*

(a)  *the wholesale and retail prices and market share of NEV increase, while those of FV decrease. Furthermore, more consumers switch to NEV from FV than the other way round.*

(b)  *Second-period profits from $R_H$ increase and profits from $R_L$ decrease.*

This result is also intuitive: consumers' concerns about the environment increase the marginal profits and market share of new energy vehicle retailers, decreasing those of fuel vehicle retailers. Consumers who are more concerned about the environment are more willing to switch to new energy vehicles and less willing to switch to fuel vehicles. Thus, new energy vehicle firms have more poaching power than fuel vehicle firms do. Furthermore, there were no constraints on this result.

*Period 1.* Using the same argument as Section 4.1, one obtains that

$$\theta_1 s_L - p_1^L + [\theta_1 s_H + \lambda(s_H - s_L) - p_N^{H*}] = \theta_1 s_H - p_1^H + [\theta_1 s_L - \lambda(s_H - s_L) - p_N^{L*}]. \tag{28}$$

$\theta_1^*$ is determined by the equation $p_1^H - p_1^L + 2\lambda(s_H - s_L) = p_N^{H*}(\theta_1^*) - p_N^{L*}(\theta_1^*)$. Retailers' payoff functions in period 1 are $\Pi_R^L(p_1^L) = (p_1^L - w_1^L)F(\theta_1^*) + \Pi_{R2}^{L*}$ and $\Pi_R^H(p_1^H) =$

$(p_1^H - w_1^H) [1 - F(\theta_1^*)] + \Pi_{R2}^{H*}$. Manufacturers' payoff functions in period 1 are $\Pi_M^L (w_1^L) = w_1^L F(\theta_1^*) + \Pi_{M2}^{L*}$ and $\Pi_M^H (w_1^H) = (w_1^H - c_H) [1 - F(\theta_1^*)] + \Pi_{M2}^{H*}$.

## 5. Analysis and Results

Several important and interesting findings are obtained by comparing the equilibrium outcomes across scenarios. This paper derives, primarily, the impact of consumers' environmental concerns on overall supply chain efficiency, consumer surplus, and social welfare. In addition, the effects of green innovation on NEV and overall supply chain efficiency are examined.

*5.1. Effects of Environmental Concerns on BBP Strategies*

**Proposition 3.** (***Prices, market share, and profits in Period 1.***) *As customers are more concerned about the environment in the case of retail BBP and wholesale-and-retail BBP (i.e., as $\lambda$ increases),*

(a)  *the first-period market share of the FV increases, and that of the NEV decreases.*
(b)  *The retail price $p_1^{L*}$ charged by $R_L$ and the wholesale price $w_1^{L*}$ charged by $M_L$ in period 1 become higher, and $p_1^{H*}$ charged by $R_H$ and $w_1^{H*}$ charged by $M_H$ become lower.*
(c)  *$R_L$'s and $M_L$'s first-period profits and total profits over the two periods increase, and the first-period profits of $R_H$ and $M_H$ decrease.*

Consumers are forward-looking and they anticipate a change in utility in the second period. They prefer to be better, i.e., from $L$ to $H$, but are less willing to be worse, i.e., from $H$ to $L$. Therefore, when consumers are more concerned about environment, they are more inclined to purchase $L$ in the first period. It leads to a higher retail price and wholesale price for the FV and lower retail price and wholesale price for the new energy vehicles in the first period.

The marginal consumer in the first period is indifferent between purchasing $L$ in period 1 and $H$ in period 2 and purchasing $H$ in period 1 and $L$ in period 2. The utility has added an extra utility of $\lambda(s_H - s_L)$, when the consumer switches to $H$ from $L$. In contrast, the utility is lowered by $\lambda(s_H - s_L)$, when the consumer switches to $L$ from $H$. As $\lambda$ increases, the marginal consumer $\theta_1$ moves to the right, i.e., the market share of Product $L$ increases and that of $H$ decreases in the first period.

In consequence, as consumers pay more attention to environment, the fuel vehicle manufacturer and retailer earn higher profits in period 1 and obtain more total profits over two periods. The increase in profits of the fuel vehicle firms in the first period exceeds the decrease in the second period. Hence, the efficiency of the fuel vehicle supply chain benefits from consumers' environment concern. Furthermore, the profits of the new energy vehicle manufacturer and retailer in period 1 decrease as $\lambda$ increases. The change in total profits of the new energy vehicle firms depends on the trade-off between the change in profits in period 1 and period 2.

**Proposition 4.** *(a) The profits of the NEV manufacturer and retailer, and FV manufacturer in the retail BBP scenario are lower than those without BBP. If consumers pay much attention to the environment, the FV retailer's profit is higher than that in the benchmark case. (b) The profits of each supply chain member in the case of wholesale-and-retail BBP are higher than those without BBP.*

In the case of wholesale-and-retail BBP, each supply chain member earns more than in the benchmark case. BBP also intensifies competition in the second period, leading to lower retail and wholesale prices than in the benchmark case. However, this mitigates competition in the first period, resulting in higher retail and wholesale prices than in the benchmark case. However, this is not the case for retail BBP. Table 2 presents a comparison of profits between the BBP and benchmark cases.

**Table 2.** Comparisons of profits between BBP cases and benchmark case.

|  | **Benchmark Case** | **Retail BBP** | **Wholesale-and-Retail BBP** |
|---|---|---|---|
| Manufacturers | $\Pi_M^{L^0*}$ <br> $\Pi_M^{H^0*}$ | $\Pi_M^{L*} < \Pi_M^{L^0*}$ <br> $\Pi_M^{H*} < \Pi_M^{H^0*}$ | $\Pi_M^{L*} > \Pi_M^{L^0*}$ <br> $\Pi_M^{H*} > \Pi_M^{H^0*}$ |
| Retailers | $\Pi_R^{L^0*}$ <br> $\Pi_R^{H^0*}$ | $\Pi_R^{L*} > \Pi_R^{L^0*} (\lambda' < \lambda < \overline{\lambda})$ <br> $\Pi_R^{H*} < \Pi_R^{H^0*}$ | $\Pi_R^{L*} > \Pi_R^{L^0*}$ <br> $\Pi_R^{H*} > \Pi_R^{H^0*}$ |

*5.2. Effects of Environmental Concerns on Supply Chain Efficiency*

Now analyze the impact of consumers' environmental concerns on supply chain efficiency.

**Proposition 5.** *If consumers are more concerned about the environment, supply chain efficiency increases with* $\lambda$ *in both BBP cases.*

Appendix B provides proofs. If $\lambda$ is sufficiently large, new energy vehicle firms demonstrate significant poaching power in the second period and obtain higher profits. When the increase in profits in the second period is greater than the decrease in profits in the first period, the total profits increase. Consequently, a firms' total profits in the supply chain increase as customers' concerns about the environment increase. Figure 5 shows the relationship between the total profits of the supply chain and $\lambda$.

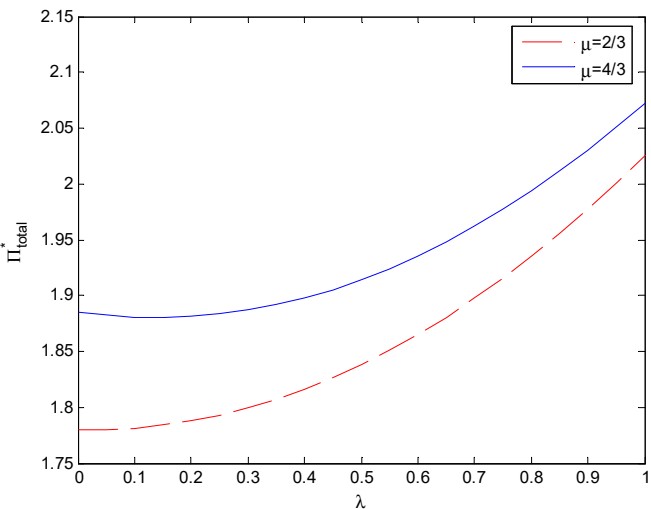

**Figure 5.** Total profits of the supply chain changing with $\lambda$ in the case of retail BBP.

*5.3. Effects of Green Innovation on Supply Chain Efficiency*

Having determined the impact of consumer environmental concerns on supply chain efficiency, this paper also explores the effect of green innovation on supply chain efficiency and how it affects NEV and overall supply chain efficiency.

**Proposition 6.** *In all cases, green innovation can improve NEV, FV, and overall supply chain efficiency.*

This conclusion seems counterintuitive; it is generally believed that green innovation will definitely improve the efficiency of the new energy vehicle supply chain, but perhaps not that of the fuel vehicle supply chain. Consequently, green innovation improves the overall supply chain efficiency. Specifically, new energy vehicle firms' green innovation has raised NEV prices, while fuel vehicle firms, as competitors, have room for price increases.

Therefore, green innovation not only benefits new energy vehicle firms, but also increases fuel vehicle firms' revenue.

*5.4. Effects of Environmental Concerns on Welfare*

**Proposition 7.** *(a) If NEV firms are relatively efficient (i.e., $0 < \mu < \hat{\mu}$), consumer surplus increases with $\lambda$ in both BBP cases. (b) If consumers are fully or less concerned about the environment and NEV firms are relatively efficient (i.e., $\widetilde{\lambda} \leq \lambda < \overline{\lambda}$ or $0 \leq \lambda < \widetilde{\lambda}, 0 \leq \mu < \widetilde{\mu}$ ), social welfare increases with $\lambda$ in both BBP cases. (c) If consumers are sufficiently concerned about the environment, NEV firms become more efficient, and a win–win–win scenario occurs for firms, consumers, and social welfare.*

Appendix B provides the proof. Because BBP intensifies competition in the second period and softens competition in the first period, all prices in the second period are lower than those in the first period. Hence, the paper focuses on prices in the first period. Furthermore, if $\mu$ is sufficiently small, new energy vehicle firms are more efficient than fuel vehicle firms; hence, the former earn greater marginal profits than the latter. Thus, new energy vehicles' prices play an important role in consumer surplus. The retail price of a new energy vehicle in the first period decreases with $\lambda$. Therefore, the consumer surplus increases with $\lambda$ if $\mu$ is sufficiently small. Furthermore, higher consumer surplus and supply chain efficiency lead to greater social welfare. Hence, when consumers are fully concerned about the environment and new energy vehicle firms are more efficient, a win–win–win scenario occurs for firms, consumers, and social welfare. In other words, it is a win–win–win situation when consumers obtain more utility when switching to new energy vehicles, and new energy vehicles are more efficient than fuel vehicles. Figure 6 shows a win–win–win scenario for the supply chain, consumer surplus, and social welfare. The solid blue line with asterisks indicates the upper limit of $\mu$, which means $\mu$ is small and the new energy vehicle firms are more efficient than fuel vehicle firms. The red dotted line represents the lower limit of $\lambda$, meaning that consumers' environmental concerns do not fall below this value. The black dotted line represents the upper limit of $\lambda$, which ensures that all equilibrium results are within reasonable range.

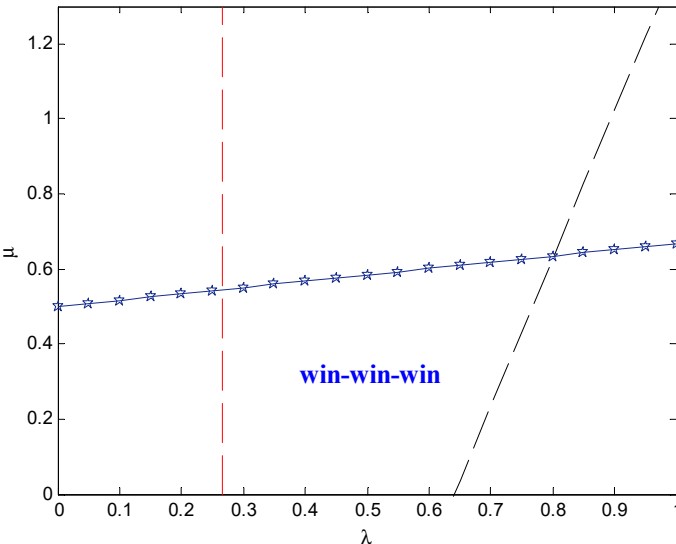

**Figure 6.** A win–win–win scenario.

## 6. Extensions

In the main models, each manufacturer produces only one type of vehicle; however, in practice, manufacturers usually produce more than one type of vehicle. Thus, a two-vehicle duopoly wherein each manufacturer produces both FV and NEV is considered. While

consumers and firms do not have discounted payoffs during the second period in the main model, the extension considers consumers' and firms' discounted payoffs in the second period.

### 6.1. Case of a Two-Vehicle Duopoly

In practice, one manufacturer usually produces a variety of vehicles. For example, Mercedes-Benz, BMW, and BYD Company in China produce both fuel vehicles and new energy vehicles. Now, we consider BBP in a two-vehicle duopoly. Each of two symmetric manufacturers $M_A$ and $M_B$ offers two types of vehicles, a fuel vehicle and a new energy vehicle, through independent retailers $R_A$ and $R_B$, respectively, to consumers in two periods, as shown in Figure 7. Assume that the two firms are situated at locations 0 and 1 on a hoteling line of length 1. Consumers' preference for the quality of the vehicle $\theta$ is also uniformly distributed on $[0,1]$. Thus, consumers are uniformly distributed on a two-dimensional plane of $[0,1] \times [0,1]$ as shown in Figure 8. In period 1, the two manufacturers set wholesale prices, $w_{i1}^L$ and $w_{i1}^H$ ($i = A, B$), for the two vehicles, respectively. Then, the two retailers set retail prices $p_{i1}^L$ and $p_{i1}^H$. In period 2, the two manufacturers set wholesale prices $w_{i2}^L$ and $w_{i2}^H$ for the two vehicles, and then the two retailers set repeat prices $p_{iO}^L$ and $p_{iO}^H$ and poach prices $p_{iN}^L$ and $p_{iN}^H$, respectively.

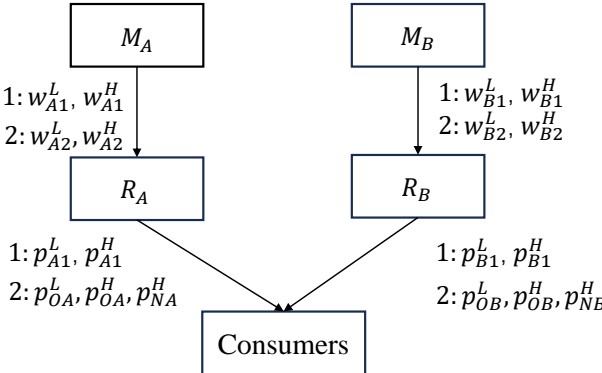

**Figure 7.** The structure of the supply chain.

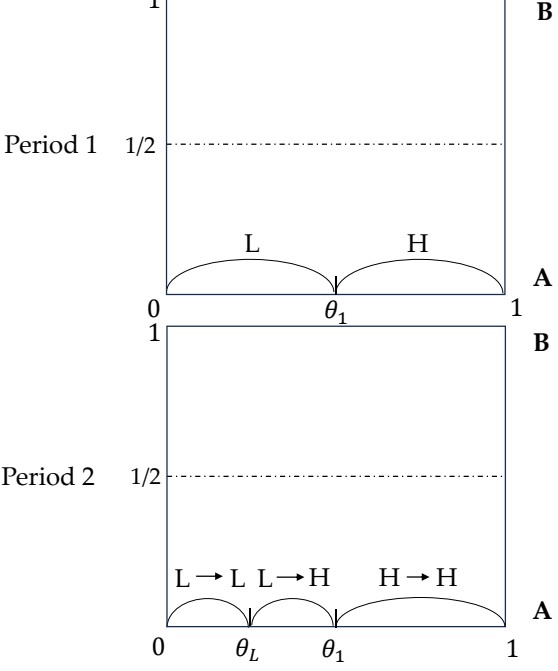

**Figure 8.** The market segmentation in a two-vehicle duopoly.

**Lemma 1.** *(a) If the FV is sufficiently efficient, BBP should not be adopted in a two-vehicle duopoly. (b) If the NEV is sufficiently efficient, some FV consumers in period 1 are induced to switch to NEV in period 2; however, no NEV consumers in period 1 are induced to switch to FVs in period 2.*

The second result of Lemma 1 is shown in Figure 7. If the fuel vehicle is sufficiently efficient in a two-vehicle duopoly, BBP leads to a contradiction. Hence, it should not be used when the manufacturer offers both types of vehicles. If the new energy vehicle is sufficiently efficient, the firm should induce consumers purchasing fuel vehicles to switch to new energy vehicles. For example, some automobile retailers recommend new energy vehicles to their customers.

*6.2. Discounted Payoffs in the Second Period*

Some extant studies have discussed cases wherein consumer discount rates differ from firm discount rates [22,48]. Another study discusses the same discount rate [21]. Here, assume that consumers have the same discount factor as firms. The discount factor is denoted by $\delta$.

For BBP, the second period analysis remained identical to that of the main model. The marginal consumer in the first period is identified as

$$V + \theta_1 s_L - p_1^L + \delta\left(V + \theta_1 s_H + \lambda(s_H - s_L) - p_N^H\right) = V + \theta_1 s_H - p_1^H + \delta\left(V + \theta_1 s_L - \lambda(s_H - s_L) - p_N^L\right). \quad (29)$$

Retailers' first-period profit functions are

$$\Pi_R^L\left(p_1^L\right) = \left(p_1^L - w_1^L\right)\theta_1 + \delta\Pi_{R2}^{L*}. \quad (30)$$

$$\Pi_R^H\left(p_1^H\right) = \left(p_1^H - w_1^H\right)(1 - \theta_1) + \delta\Pi_{R2}^{H*}. \quad (31)$$

Manufacturers' first-period profit functions are as follows:

$$\Pi_M^L\left(w_1^L\right) = w_1^L\theta_1 + \delta\Pi_{M2}^{L*}. \quad (32)$$

$$\Pi_M^H\left(w_1^H\right) = \left(w_1^H - c_H\right)(1 - \theta_1) + \delta\Pi_{M2}^{H*}. \quad (33)$$

Similar results as the main model are shown as follows.

**Lemma 2.** *The profits of each supply chain player in the case of wholesale-and-retail BBP are higher than those in the case of retail BBP. Hence, the supply chain efficiency in the former case is greater than that in the latter. However, the consumer surplus and social welfare in the former case are lower than those in the latter.*

When considering the discounted payoffs of consumers and firms in the second period, the results are similar to those of the main model. The total profits of the supply chain when both manufacturers and retailers adopt BBP are higher than when only retailers adopt BBP. However, higher prices in the former case cause consumers to obtain lower surpluses.

In addition, when the discount factor is considered in the second period, the paper also finds that the second-period profits for new energy vehicle firms increase with $\lambda$ while those for fuel vehicle firms decrease with $\lambda$. However, the opposite is true in the first period. Because firms discount their profits in the second period, new energy vehicle manufacturers and retailers experience a lower increase in profits as $\lambda$ increases. In particular, when the discount factor is small, the total profits of the new energy vehicle manufacturer and retailer over the two periods increase with $\lambda$ if consumers pay sufficient attention to the environment, and the new energy vehicle is relatively efficient.

This study has several limitations. First, we assume that consumers' quality preferences follow a uniform distribution. This reduces the robustness of the model. Further,

for technical simplicity, assume that the environmental value of a new energy vehicle is reflected in its quality. These two problems should be addressed in future research.

## 7. Empirical Analysis

Through analytical modeling, this paper finds that consumers' environmental concerns can improve the market competitiveness of NEV firms, which further increases supply chain efficiency, and that green innovation can enhance the efficiency of the NEV supply chain. This section further validates the main results of this study by using empirical methods to demonstrate the robustness of the analytical model.

### 7.1. Research Methodology

#### 7.1.1. Sampling and Data Collection

This study considers new energy vehicle firms as research objects. Because the new energy vehicle industry cannot be used as a standard for classification, this study considers the main business of firms as the standard for classification. The main business includes raw materials, upstream and downstream, car manufacturing, assembly structures, and other related matters in the automobile industry. This paper considers firms involving "NEV" business as samples. The selected samples were processed as follows: firms with PT, ST, and *ST were removed; enterprises with missing data on major variables were eliminated; and insolvent enterprises were excluded. After screening, data from 463 A-share-listed new energy vehicle companies from 2010 to 2021 were obtained. To avoid the impact of extreme values on the results, all continuous variables were winsorized at the 1% and 99% levels.

#### 7.1.2. Variables and Measurements

(1)     Dependent variable: Enterprise Performance (TobinQ)

In this study, TobinQ was used to measure the performance of NEV enterprises. In the robustness test, return on total assets (ROA) and return on equity (ROE) were used as surrogate variables to measure enterprise performance for regression analysis.

(2)     Independent variable: Public Environmental Concerns (PECs)

With the continuous development of the Internet and the rapid circulation of data, an increasing number of people rely on search engines to obtain the desired information, and a relevant search index can reflect the public's concerns and information demands. According to the measurement method of PEC in the existing literature, the search engine of Baidu uses "smog" and "environmental pollution" as keywords to obtain the total search volume index of PC and mobile terminal of the provinces. Then, the paper matched the corresponding province where the listed company is located to determine the PEC data.

(3)     Intermediary Variables

Green Innovation (lnGI): The main actors involved in green innovation are mainly new energy vehicle manufacturers. Green innovation refers specifically to green technological innovation, from the drive motor, electronic control system, battery technology and other aspects to reduce energy consumption and emissions. New energy vehicle manufacturers improve the level of green innovation by increasing R&D investment in green technologies. One of the mediating variables in this study is green innovation performance. Referring to previous relevant studies ([33]), we use the natural logarithm of the number of green patent applications plus one to measure the green innovation level. Patent data for green innovation were obtained from the China Research Data Service Platform (CNRDS).

Market Competitiveness (EPCM): Market competitiveness was another intermediate variable. The relative Lerner index is used as a proxy index, and the Lerner index, adjusted by the industry average, is used to measure the competitive position of enterprises in the market. Specifically, the difference between the Lerner index of a single company and that of the average sales-weighted index of all listed companies in the same industry was used to obtain an index to measure the competitiveness of a company.

(4)    Control variables

Select the nature of property rights (SOE), firm size (Size), solvency (Lev), the top 10 shareholders' shareholding ratio (Top10), board size (Board), management compensation (TMTPay), firm establishment age (FirmAge), and the institutional investors' shareholding ratio (INST) as control variables. The data were extracted from the China Economic and Financial Research Database (CSMAR) and the annual reports of each company, as shown in Table 3.

**Table 3.** Variable definitions.

| Variable Type | Variable Name | Variable Measurement |
|---|---|---|
| Dependent variable | Enterprise performance (TobinQ) | (Market value of tradable shares + number of non-tradable shares × net assets per share + book value of liabilities)/Total assets |
| Independent variable | Public environmental concerns (PECs) | Baidu search index with "smog" and "environmental pollution" as keywords/100 |
| Intermediary variable | Green innovation (lnGI) Market competitiveness (EPCM) | Natural logarithm of (number of green patent applications +1) Relative to the Lerner index |
| Control variable | Property right nature (SOE) Firm size (Size) Solvency (Lev) Top10 shareholders' Shareholding ratio (Top10) Board size (Board) Management compensation (TMTPay) Firm establishment age (FirmAge) Institutional investors' shareholding ratio (INST) | Dummy variable: 1 = state-owned and 0 = others The natural log of total assets Total liabilities/total assets Number of shares held by top 10 shareholders/total number of shares The natural logarithm of the number of board members The natural logarithm of total executive compensation ln(current year—year of company establishment +1) Total shares held by institutional investors/Total share capital |

### 7.2. Empirical Results and Analysis

#### 7.2.1. Descriptive Statistics Analysis

Table 4 presents the descriptive statistics of the main variables. The minimum and maximum values of TobinQ are 0.815 and 5.289, respectively, the mean and median are small, and the standard deviation is 0.819, indicating a large performance gap between the different NEV enterprises in the sample. The mean value of public environmental concern is 4.082, the median is 3.739, and the standard deviation is 1.893, indicating that the NEV industry in the sample was evenly distributed in the provinces with higher and lower public environmental concerns. The minimum value of the green innovation index is zero, indicating that some firms did not implement green innovation practices in a specific year. The minimum value of the relative Lerner index, which measures the market competitiveness of an enterprise, is $-0.345$. A negative value indicates that the corresponding enterprise has a small market share and no market power in the industry. A high relative Lerner index indicates that an enterprise has a large market share and a certain market power in the industry.

**Table 4.** Summary statistics.

| VarName | Obs | Mean | SD | Min | Median | Max |
|---|---|---|---|---|---|---|
| TobinQ | 3178 | 1.821 | 0.819 | 0.815 | 1.559 | 5.289 |
| PEC | 3178 | 4.082 | 1.893 | 0.577 | 3.739 | 8.512 |
| lnGI | 3178 | 1.175 | 1.376 | 0.000 | 0.693 | 7.223 |
| EPCM | 3178 | 0.017 | 0.085 | −0.345 | 0.014 | 0.204 |
| SOE | 3178 | 0.250 | 0.433 | 0.000 | 0.000 | 1.000 |
| Size | 3178 | 22.132 | 1.133 | 19.765 | 21.972 | 24.678 |
| Lev | 3178 | 0.421 | 0.175 | 0.056 | 0.424 | 0.751 |
| Top10 | 3178 | 59.011 | 13.866 | 20.397 | 59.763 | 82.165 |
| Board | 3178 | 2.115 | 0.192 | 1.386 | 2.197 | 2.890 |
| TMTPay | 3178 | 15.298 | 0.680 | 13.095 | 15.282 | 16.915 |
| FirmAge | 3178 | 2.849 | 0.345 | 1.386 | 2.890 | 3.638 |
| INST | 3178 | 41.882 | 25.927 | 0.000 | 43.170 | 157.098 |

### 7.2.2. Benchmark Regression Results

In this study, the OLS model was used for sample regression, and the results are shown in Table 5. Column (1) shows the regression results that do not control for other variables and fixed effects; the regression coefficient is 0.087, which is significantly positive at the 1% level. Column (2) shows the regression results with the control variables added, based on Column (1). The regression coefficient is 0.089, which is significantly positive at the 1% level. In column (3), not only are control variables added, but time and province fixed effects are also controlled for, and the regression coefficient is 0.035, which is significantly positive at the 10% level. The above analysis indicates that PECs play a positive role in promoting the performance of NEV enterprises.

**Table 5.** Benchmark regression results.

| Variables | (1) TobinQ | (2) TobinQ | (3) TobinQ |
|---|---|---|---|
| PEC | 0.087 *** | 0.089 *** | 0.035 * |
| | (11.516) | (12.882) | (1.866) |
| SOE | | −0.158 *** | −0.200 *** |
| | | (−4.494) | (−5.686) |
| Size | | −0.310 *** | −0.311 *** |
| | | (−17.275) | (−18.711) |
| Lev | | −0.419 *** | −0.382 *** |
| | | (−4.494) | (−4.468) |
| Top10 | | −0.016 *** | −0.013 *** |
| | | (−14.523) | (−12.577) |
| Board | | −0.364 *** | −0.261 *** |
| | | (−5.076) | (−3.818) |
| TMTPay | | 0.169 *** | 0.136 *** |
| | | (7.133) | (5.577) |
| FirmAge | | 0.079 ** | −0.023 |
| | | (2.005) | (−0.543) |
| INST | | 0.010 *** | 0.010 *** |
| | | (15.793) | (16.199) |
| _cons | 1.468 *** | 7.018 *** | 7.370 *** |
| | (43.365) | (18.551) | (19.267) |
| FE_Year | No | No | Yes |
| FE_Province | No | No | Yes |
| N | 3178 | 3178 | 3178 |
| adj. $R^2$ | 0.040 | 0.208 | 0.364 |

Note: $t$ statistics in parentheses; * $p < 0.1$, ** $p < 0.05$, *** $p < 0.01$.

### 7.2.3. Analysis of the Mediation Effect Test

Public environmental concerns play a positive role in promoting the performance of NEV companies. Do green innovation and market competitiveness have intermediary effects? A stepwise regression method was used to assess this relationship.

Columns (1) and (2) of Table 6 show the regression results when considering green innovation as the intermediary variable, and columns (3) and (4) show the regression results when considering market competitiveness as the intermediary variable. The results in Columns (1) and (2) show that the positive impact of PECs on green innovation is significant at the 1% level and that the positive impact of green innovation on corporate performance is also significant at the 1% level, indicating that green innovation plays a mediating role in the positive correlation between PECs and corporate performance. From the results in columns (3) and (4), the paper finds that the positive impact of PECs on the market competitiveness of enterprises is significant at the 10% level.

In turn, the positive impact of market competitiveness on enterprise performance is significant at the 1% level, indicating that market competitiveness also plays an intermediary role in the positive correlation between PECs and enterprise performance. The above analysis shows that the PECs can improve the performance level of NEV enterprises by improving their green innovation level and market competitiveness.

**Table 6.** Mediating effect test results.

| Variables | (1) lnGI | (2) TobinQ | (3) EPCM | (4) TobinQ |
|---|---|---|---|---|
| PEC | 0.072 *** | | 0.002 * | |
| | (4.254) | | (1.799) | |
| lnGI | | 0.068 *** | | |
| | | (6.031) | | |
| EPCM | | | | 2.030 *** |
| | | | | (12.567) |
| SOE | 0.160 *** | −0.105 *** | −0.038 *** | −0.017 |
| | (3.106) | (−3.212) | (−10.680) | (−0.520) |
| Size | 0.614 *** | −0.356 *** | −0.000 | −0.314 *** |
| | (23.374) | (−19.712) | (−0.011) | (−19.166) |
| Lev | 0.449 *** | −0.272 *** | −0.171 *** | 0.105 |
| | (3.385) | (−3.222) | (−18.620) | (1.205) |
| Top10 | 0.000 | −0.012 *** | 0.001 *** | −0.014 *** |
| | (0.196) | (−11.891) | (6.340) | (−13.431) |
| Board | 0.729 *** | −0.225 *** | −0.034 *** | −0.107 * |
| | (7.017) | (−3.391) | (−4.691) | (−1.650) |
| TMTPay | 0.130 *** | 0.111 *** | 0.018 *** | 0.083 *** |
| | (3.590) | (4.845) | (7.326) | (3.663) |
| FirmAge | −0.143 ** | −0.024 | −0.014 *** | −0.006 |
| | (−2.231) | (−0.589) | (−3.074) | (−0.157) |
| INST | −0.002 ** | 0.010 *** | 0.000 *** | 0.010 *** |
| | (−2.060) | (17.336) | (3.726) | (16.578) |
| _cons | −13.904 *** | 8.095 *** | −0.148 * | 7.444 *** |
| | (−12.090) | (10.808) | (−1.860) | (10.353) |
| FE | Yes | Yes | Yes | Yes |
| *N* | 3178 | 3178 | 3178 | 3178 |
| adj. $R^2$ | 0.469 | 0.392 | 0.332 | 0.415 |

Note: *t* statistics in parentheses; * $p < 0.1$, ** $p < 0.05$, *** $p < 0.01$.

### 7.2.4. Robustness Tests

(1)  Replace the dependent variable

Return on total assets (ROA) and return on equity (ROE) were used as proxy variables for firm performance. Columns (1) and (2) of Table 7 show the regression results. According to the regression results, after changing the dependent variables, the positive correlation between PECs and enterprise performance remained significant, and the above results did not change.

**Table 7.** Robustness test results.

| Variables | (1) ROA | (2) ROE | (3) TobinQ | (4) TobinQ |
|---|---|---|---|---|
| PEC | 0.004 *** | 0.009 *** | | 0.051 * |
| | (2.588) | (2.774) | | (1.932) |
| PEC$_{t-1}$ | | | 0.043 ** | |
| | | | (2.152) | |
| _cons | −0.279 *** | −0.655 *** | 8.328 *** | 8.868 *** |
| | (−9.700) | (−10.194) | (18.101) | (17.587) |
| Controls | Yes | Yes | Yes | Yes |
| FE | Yes | Yes | Yes | Yes |
| *N* | 2941 | 2941 | 2635 | 2565 |
| adj. $R^2$ | 0.256 | 0.166 | 0.396 | 0.348 |

Note: *t* statistics in parentheses; * $p < 0.1$, ** $p < 0.05$, *** $p < 0.01$.

(2)  Change in time window

There may be a lag in the impact of PECs on enterprise performance, which is achieved by adding PEC with a one-period lag as an independent variable, which can also solve the endogeneity problem to a certain extent. The regression results are shown in Column (3) of Table 5. According to the results, a PEC with a one-period lag has a significantly positive impact on enterprise performance, which enhances the credibility of the results.

(3)　　Change in sample time

The subsidy policy for the promotion and application of new energy vehicles issued jointly by the relevant departments of China's national ministries and commissions in 2013 provided important support for the development of new energy vehicle enterprises. The new energy vehicle industry has developed rapidly. Therefore, using 2013 as the time node, the samples after this node were selected for re-regression, and the regression results are shown in Column (4). The regression results show that the regression coefficient of PECs is still significant, which verifies the robustness of the results.

### 7.2.5. Impact of Environmental Concerns on the Economy

In order to explore the relationship between consumers' environmental concerns and economic development, this paper takes 30 provinces in China from 2011 to 2022 as research objects for further analysis. The independent variable is still public environmental concern (PEC), and the dependent variable is the level of real GDP per capita (Pgdp). Control variables include the level of infrastructure, the degree of openness to the outside world, the degree of government intervention, and the level of human capital. Columns (1) and (2) of Table 8 show the regression results. Regardless of whether control variables are added, the regression coefficients of PEC to Pgdp are significantly positive, indicating that the improvement in public environmental concern can promote regional economic development.

**Table 8.** Analysis results.

| Variables | (1)<br>Pgdp | (2)<br>Pgdp |
|---|---|---|
| PEC | 12.732 *** | 2.680 *** |
| | (10.457) | (3.437) |
| _cons | 7259.212 *** | 1895.638 *** |
| | (17.094) | (2.736) |
| Controls | No | Yes |
| N | 372 | 372 |
| adj. $R^2$ | 0.226 | 0.773 |

Note: $t$ statistics in parentheses; *** $p < 0.01$.

In summary, the paper used empirical analysis to verify the main conclusions obtained from the analytical model in the previous sections. The results of baseline regression and the robustness test indicate that PEC has a significant positive impact on the performance improvement of new energy vehicle firms. As an external pressure, the public demand for environmental protection encourages new energy vehicle firms to pay more attention to the driving role of investment in environmental protection projects on performance. At the same time, this is also a huge development opportunity for the new energy vehicle field. By fully tapping the potential of new energy and actively responding to public demands for environmental protection, new energy automobile firms can achieve steady improvements in corporate performance.

In addition, the mediation effect test proves that market competitiveness and green innovation play a significant intermediary role, which provides ideas for new energy vehicle firms to identify the key focus points for improving performance under the background of increasing public environmental concern. New energy vehicle firms give full play to their own new energy advantages to open up new market opportunities, and continuously improve the environmental performance of products and services through green innovation, which can further improve the performance level. The empirical conclusions provide practical enlightenment for encouraging new energy vehicle firms to seize the opportunity of environmental protection and improve economic benefits and competitiveness by actively adopting effective green behaviors and environmental protection inputs.

## 8. Discussions and Conclusions

### 8.1. Discussions

This section compares the results of the related literature with those of this paper. Regarding the competition between new energy vehicles and traditional vehicles, Zhao et al. (2022) [9] focus on the conditions for the supply chain coordination of new energy vehicle and traditional vehicle supply chains. The impact of the degree of sophistication of charging facilities on the pricing, demand and supply chain efficiency of the two types of vehicles is further investigated. They did not take into account consumers' environmental concerns and green innovations. In addition, Li et al. (2020) [1] found that the battery recycling rate is the most crucial element influencing a new energy vehicle manufacturer's competitive status. Differently, this paper finds that consumers' environmental concern and the green innovation of new energy vehicle firms affects BBP strategies, which in turn impacts the competitiveness and supply chain efficiency of new energy vehicles.

This paper first draws on previous studies on BBP strategies in vertically differentiated competition, and it examines BBP competitive strategies in green supply chains. The paper obtains similar findings to the previous study that wholesale-and-retail BBP strategies are superior to retail BBP strategies. Different from previous research, this paper considers the impact of consumers' environmental concerns and green innovations on the competition between new energy and fuel vehicles. The impacts of consumers' environmental concerns and green innovations on the supply chain efficiency, consumer surplus, and social welfare are further obtained.

Recently published related work is of great value for green supply chain development. For example, Heydari et al. (2021) [11] analyze the green channel coordination problem in a two-echelon supply chain wherein the retailer decides on the selling price, while the manufacturer regulates the green quality of the product. They initiate the channel coordination and establish a win–win outcome for both parties. The green quality of the product is a key issue that deserves to be studied. Cheng and Fan (2021) [49] examine the competition and coopetition for a fuel vehicle automaker and a rival new energy vehicle automaker under the dual-credit policy, and obtain the impact of the credit coefficient, credit equilibrium, and NEV substitutability on both parties' production decisions and profits. This study focuses on the impact of a dual-credit policy. Subsequently, Liao et al. (2022) [12] investigate the role of governmental policy in competition between traditional fuel and new energy vehicles, and analyze the impacts of different regulation intensities on the promotion effect of NEVs. In addition, Liu et al. (2023) [50] explore how blockchain adoption affects the duopoly competition between green and non-green products, and obtain that blockchain adoption decreases price competition between green and non-green products, causing both the green and non-green product manufacturers to raise prices. The innovation of this paper is the consideration of blockchain traceability techniques in green supply chain competition.

The above literature has an excellent research perspective regarding the green supply chain research issue, which is worthwhile to follow. Scholars have explored technologies or policies that are favorable to the development of green supply chains. This paper discusses the role of promoting the development of a new energy vehicle supply chain mainly from the consumer level. Therefore, it is necessary to examine this important issue from the aspects of both internal and external factors that promote the development of green supply chains.

When consumers are not environmentally conscious, $\lambda = 0$. By analyzing the analytical results in this paper, we can obtain that the market competitiveness of new energy vehicles will decline in the second period. In other words, consumers' lack of concern for the environment is detrimental to market competition for new energy vehicles. In addition, overall supply chain efficiency and consumer surplus will decrease, and therefore a win–win–win situation cannot be realized.

### 8.2. Conclusions

In response to the research questions posed above, this paper has obtained the following conclusions through analytical methods: (i) Firstly, consumers' environmental concerns improve the competitiveness of new energy vehicle firms in the second period. (ii) Wholesale-and-retail BBP strategies benefit all players of the supply chain, but this is not the case for retail BBP strategies. (iii) If consumers care about the environment sufficiently, supply chain efficiency is improved in cases of retail as well as wholesale-and-retail BBP strategies. (iv) Green innovation improves new energy vehicles, fuel vehicles, as well as overall supply chain efficiency. (v) If consumers are sufficiently concerned about the environment and new energy vehicle firms are more efficient, a win–win–win scenario for firms, consumers, and social welfare occurs in the two BBP strategies. (vi) An improvement in consumers' environmental concern can promote regional economic development. Furthermore, the hypotheses and conclusions that were and were not confirmed using empirical methods are listed in Table 9.

**Table 9.** Confirmed and unconfirmed hypotheses and conclusions using empirical methods.

| Confirmed Hypotheses | Unconfirmed Hypotheses and Conclusions |
| --- | --- |
| Consumers' environmental concerns increase the performance of NEV firms by increasing the level of green innovation | Effects of consumers' environmental concerns on FV supply chain efficiency |
| Consumers' environmental concerns increase the performance of NEV firms by increasing the competitiveness of NEV firms | Effects of consumers' environmental concerns and green innovation on social welfare |
| Consumers' environmental concerns improve economic development. | |

This study examines how consumers' environmental concerns and green innovation affect competition between new energy vehicles and fuel vehicles in a vertically differentiated duopoly. Meanwhile, new energy vehicle and fuel vehicle firms use BBP strategies to compete for the market. The theoretical value of this paper is to consider the impact of consumers' environmental concerns and green innovations on new energy vehicle and fuel vehicle firms' competitiveness, and to investigate the BBP strategies in a green supply chain, which is a theoretical contribution. Moreover, the managerial implication at the practical level is that new energy vehicle firms should emphasize an R&D investment in green innovative technologies, and pay attention to the environmental concern of consumers in the potential market, which is conducive to their competitive performance with fuel vehicle firms. In addition, competing firms need to choose their BBP strategies cautiously, as BBP strategies are only designed to compete for markets and do not necessarily benefit supply chain efficiency.

In addition, this study has some potential and extensibility. Firstly, the theoretical model is used to portray the real problem, and then the conclusions of the analytical model are verified by empirical methods. The possible innovations and future research directions include competition between multiple stakeholders that produce both new energy and fuel vehicles, and the further consideration of carbon emissions from new energy vehicle and fuel vehicle firms. Another future research issue is the impact of government policies on new energy vehicles and their development.

This study is feasible in practice to a certain extent. The validation of the empirical part uses readily available data from new energy vehicle firms. After screening, data from 463 A-share-listed new energy vehicle firms from 2010 to 2021 were obtained. To avoid the impact of extreme values on the results, all continuous variables were winsorized at the 1% and 99% levels. Some assumptions about the parameters need to be used in the analytical modeling part. The application of this research lies at the firm level and is mainly applied to the competition between new energy vehicle and traditional fuel vehicle firms. This study puts forward suggestions for improving the competitiveness of new energy vehicle

firms, which is conducive to promoting the development of the new energy automobile industry.

**Author Contributions:** S.C.: Conceptualization, Formal Analysis, Methodology, Writing—Original Draft. G.L.: Formal Analysis—Review and Editing. All authors have read and agreed to the published version of the manuscript.

**Funding:** This research was funded by the National Natural Science Foundation of China [grant number 72104172], the Research Project Supported by Shanxi Scholarship Council of China [grant number 2023-037], and the School Fund of Taiyuan University of Technology [grant number 2022QN126].

**Institutional Review Board Statement:** Not applicable.

**Informed Consent Statement:** Not applicable.

**Data Availability Statement:** The data are available on request from the corresponding author.

**Conflicts of Interest:** The authors declare no conflicts of interest.

## Appendix A

The equilibrium results without BBP are summarized in Table A1.

**Table A1.** Equilibrium results without BBP.

| Variables | Results |
|---|---|
| $w_1^{L^0*}$ | $w_1^{L^0*} = (s_\Delta/3)(4+\mu)$ |
| $w_1^{H^0*}$ | $w_1^{H^0*} = (s_\Delta/3)(5+2\mu)$ |
| $p_1^{L^0*}$ | $p_1^{L^0*} = (4s_\Delta/9)(4+\mu)$ |
| $p_1^{H^0*}$ | $p_1^{H^0*} = (5s_\Delta/9)(4+\mu)$ |
| $\Pi_M^{L^0*}$ | $\Pi_M^{L^0*} = (s_\Delta/27)(4+\mu)^2$ |
| $\Pi_M^{H^0*}$ | $\Pi_M^{H^0*} = (s_\Delta/27)(5-\mu)^2$ |
| $\Pi_R^{L^0*}$ | $\Pi_R^{L^0*} = (s_\Delta/81)(4+\mu)^2$ |
| $\Pi_R^{H^0*}$ | $\Pi_R^{H^0*} = (s_\Delta/81)(5-\mu)^2$ |

The equilibrium results in the case of retail BBP are summarized in Table A2.

**Table A2.** Equilibrium results in the case of retail BBP.

| Variables | Results |
|---|---|
| $w_1^{L*}$ | $w_1^{L*} = 2s_\Delta(26,284 + 8087\lambda + 6817\mu)/44,955$ |
| $w_1^{H*}$ | $w_1^{H*} = s_\Delta(66,202 - 16,174\lambda + 31,321\mu)/44,955$ |
| $p_1^{L*}$ | $p_1^{L*} = (11,136\mu + 52,337 + 12,516\lambda)s_\Delta/29,970$ |
| $p_1^{H*}$ | $p_1^{H*} = (63,473 - 12,516\lambda + 18,834\mu)s_\Delta/29,970$ |
| $p_O^{L*}$ | $p_O^{L*} = (1888\mu + 3385 - 1564\lambda)s_\Delta/3996$ |
| $p_O^{H*}$ | $p_O^{H*} = (2108\mu + 5273 + 1564\lambda)s_\Delta/3996$ |
| $p_N^{L*}$ | $p_N^{L*} = (7424\mu + 14,603 - 11,636\lambda)s_\Delta/19,980$ |
| $p_N^{H*}$ | $p_N^{H*} = (12,556\mu + 22,027 + 11,636\lambda)s_\Delta/19,980$ |
| $w_2^{L*}$ | $w_2^{L*} = (3941 - 2432\lambda + 2108\mu)s_\Delta/6660$ |
| $w_2^{H*}$ | $w_2^{H*} = (6049 + 2432\lambda + 4552\mu)s_\Delta/6660$ |
| $\theta_L^*$ | $\theta_L^* = (2551 - 262\lambda + 1558\mu)/9990$ |
| $\theta_1^*$ | $\theta_1^* = (499 + 212\lambda + 112\mu)/1110$ |
| $\theta_H^*$ | $\theta_H^* = (5881 - 262\lambda + 1558\mu)/9990$ |
| $\Pi_M^{L*}$ | $\Pi_M^{L*} = \dfrac{s_\Delta\left[31{,}459{,}424\lambda^2 + 8\lambda(2{,}419{,}237 - 1{,}494{,}044\mu) + 11\left(13{,}774{,}561 + 9{,}146{,}336\mu + 1{,}767{,}184\mu^2\right)\right]}{199{,}600{,}200}$ |
| $\Pi_M^{H*}$ | $\Pi_M^{H*} = \dfrac{s_\Delta\left[31{,}459{,}424\lambda^2 - 8\lambda(925{,}193 + 1{,}494{,}044\mu) + 11\left(24{,}688{,}081 - 12{,}680{,}704\mu + 1{,}767{,}184\mu^2\right)\right]}{199{,}600{,}200}$ |
| $\Pi_R^{L*}$ | $\Pi_R^{L*} = \dfrac{s_\Delta\left[17{,}162{,}663 + 2{,}939{,}972\lambda^2 + 9{,}175{,}888\mu + 1{,}708{,}772\mu^2 + 8\lambda(388{,}936 - 82{,}457\mu)\right]}{49{,}900{,}050}$ |
| $\Pi_R^{H*}$ | $\Pi_R^{H*} = \dfrac{s_\Delta\left[28{,}047{,}323 + 2{,}939{,}972\lambda^2 - 12{,}593{,}432\mu + 1{,}708{,}772\mu^2 - 8\lambda(306{,}479 + 82{,}457\mu)\right]}{49{,}900{,}050}$ |

To achieve $\theta_L^* < \theta_1^* < \theta_H^*$, the conditions of $\mu \in (0, \overline{\mu})(\overline{\mu} = 2)$ and $\lambda \in (0, \overline{\lambda})\left(\overline{\lambda} = \frac{139+55\mu}{217}\right)$ need to be satisfied. The condition ensures the existence of the pure strategy equilibrium. The equilibrium results in the case of wholesale-and-retail BBP are summarized in Table A3.

**Table A3.** Equilibrium results in the case of wholesale-and-retail BBP.

| Variables | Results |
|---|---|
| $w_1^{L*}$ | $w_1^{L*} = s_\Delta(251{,}930 + 14{,}371\lambda + 12{,}764\mu)/45{,}684$ |
| $w_1^{H*}$ | $w_1^{H*} = s_\Delta(264{,}694 - 14{,}371\lambda + 32{,}920\mu)/45{,}684$ |
| $p_1^{L*}$ | $p_1^{L*} = s_\Delta(58{,}490 + 3435\lambda + 2964\mu)/7614$ |
| $p_1^{H*}$ | $p_1^{H*} = s_\Delta(61{,}454 - 3435\lambda + 4650\mu)/7614$ |
| $w_O^{L*}$ | $w_O^{L*} = s_\Delta(550 - 229\lambda + 310\mu)/846$ |
| $w_O^{H*}$ | $w_O^{H*} = s_\Delta(860 + 229\lambda + 536\mu)/846$ |
| $w_N^{L*}$ | $w_N^{L*} = s_\Delta(1762 - 1393\lambda + 988\mu)/3384$ |
| $w_N^{H*}$ | $w_N^{H*} = s_\Delta(2750 + 1393\lambda + 2396\mu)/3384$ |
| $p_O^{L*}$ | $p_O^{L*} = 2s_\Delta(550 - 229\lambda + 310\mu)/1269$ |
| $p_O^{H*}$ | $p_O^{H*} = s_\Delta(1720 + 458\lambda + 649\mu)/1269$ |
| $p_N^{L*}$ | $p_N^{L*} = s_\Delta(1762 - 1393\lambda + 988\mu)/2538$ |
| $p_N^{H*}$ | $p_N^{H*} = s_\Delta[1393\lambda + 50(55 + 31\mu)]/2538$ |
| $\theta_L^*$ | $\theta_L^* = (550 - 229\lambda + 310\mu)/2538$ |
| $\theta_1^*$ | $\theta_1^* = (550 + 53\lambda + 28\mu)/1128$ |
| $\theta_H^*$ | $\theta_H^* = (1678 - 229\lambda + 310\mu)/2538$ |
| $\Pi_M^{L*}$ | $\Pi_M^{L*} = \frac{s_\Delta[300{,}956{,}932 + 9{,}861{,}841\lambda^2 + 54{,}961{,}616\mu + 8{,}256{,}016\mu^2 - 76\lambda(-206{,}509 + 169{,}934\mu)]}{103{,}063{,}104}$ |
| $\Pi_M^{H*}$ | $\Pi_M^{H*} = \frac{s_\Delta[364{,}174{,}564 + 9{,}861{,}841\lambda^2 - 71{,}473{,}648\mu + 8{,}256{,}016\mu^2 - 76\lambda(36{,}575 + 169{,}934\mu)]}{103{,}063{,}104}$ |
| $\Pi_R^{L*}$ | $\Pi_R^{L*} = \frac{s_\Delta[3{,}440{,}839\lambda^2 + \lambda(8{,}418{,}628 - 4{,}142{,}744\mu) + 4(29{,}213{,}911 + 5{,}001{,}068\mu + 698{,}716\mu^2)]}{103{,}063{,}104}$ |
| $\Pi_R^{H*}$ | $\Pi_R^{H*} = \frac{s_\Delta[3{,}440{,}839\lambda^2 - 4\lambda(1{,}068{,}971 + 1{,}035{,}686\mu) + 4(34{,}913{,}695 - 6{,}398{,}500\mu + 698{,}716\mu^2)]}{103{,}063{,}104}$ |

If $\mu \in (0, \overline{\mu})$ and $\lambda \in (0, 1)$, $\theta_L^* < \theta_1^* < \theta_H^*$.

**Appendix B**

**Proof of Proposition 1.** First analyze the case of retail BBP in period 2.

(a)  $\frac{\partial p_O^{L*}}{\partial \lambda} = \frac{-1564 s_\Delta}{3996} < 0$, $\frac{\partial p_N^{L*}}{\partial \lambda} = -\frac{11{,}636 s_\Delta}{19{,}980} < 0$, $\frac{\partial p_O^{H*}}{\partial \lambda} = \frac{1564 s_\Delta}{3996} > 0$, $\frac{\partial p_N^{H*}}{\partial \lambda} = \frac{11{,}636 s_\Delta}{19{,}980} > 0$,
$\frac{\partial w_2^{L*}}{\partial \lambda} = \frac{-2432 s_\Delta}{6660} < 0$, and $\frac{\partial w_2^{H*}}{\partial \lambda} = \frac{2432 s_\Delta}{6660} > 0$.

(b)  The market segmentation of consumers switch to $H$ from $L$ is $\theta_1^* - \theta_L^* = \frac{194 + 217\lambda - 55\mu}{999}$ and $\frac{\partial(\theta_1^* - \theta_L^*)}{\partial \lambda} = \frac{217}{999} > 0$. The market segmentation of consumers' switch to $L$ from $H$ is $\theta_H^* - \theta_1^* = \frac{139 - 217\lambda + 55\mu}{999}$ and $\frac{\partial(\theta_H^* - \theta_1^*)}{\partial \lambda} = \frac{-217}{999} < 0$.

(c)  If $\mu \in [0, 2]$ and $\lambda \in [0, \overline{\lambda}]\left(\overline{\lambda} = (139 + 55\mu)/217\right)$,

$$\frac{\partial \Pi_{R2}^{L*}}{\partial \lambda} = \frac{s_\Delta(-1{,}842{,}331 + 2{,}388{,}772\lambda - 800{,}848\mu)}{24{,}950{,}025} < 0, \tag{A1}$$

$$\frac{\partial \Pi_{R2}^{H*}}{\partial \lambda} = \frac{s_\Delta(2{,}643{,}179 + 2{,}388{,}772\lambda - 800{,}848\mu)}{24{,}950{,}025} > 0, \tag{A2}$$

$$\frac{\partial \Pi_{M2}^{L*}}{\partial \lambda} = \frac{608 s_\Delta(-3941 + 2432\lambda - 2108\mu)}{8{,}316{,}675} < 0, \tag{A3}$$

$$\frac{\partial \Pi_{M2}^{H*}}{\partial \lambda} = \frac{608 s_\Delta(6049 + 2432\lambda - 2108\mu)}{8{,}316{,}675} > 0. \tag{A4}$$

□

**Proof of Proposition 2.** The proof is parallel to Proposition 1 and is omitted here. □

**Proof of Proposition 3.** (a) In the case with retail BBP, the retail price charged to new customers is $p_N^{H*} = s_\Delta(22,027 + 11,636\lambda + 12,556\mu)/19,980$, and the retail price charged to past customers is $p_O^{H*} = s_\Delta(5273 + 1564\lambda + 2108\mu)/3996$. Solve the inequation of $p_N^{H*} > p_O^{H*}$, one obtains the condition of $617/972 < \mu < \overline{\mu}$ and $(241 - 112\mu)/212 < \lambda < \overline{\lambda}$, where $\overline{\lambda} = (139 + 55\mu)/217$, $\overline{\mu} = 2$. (b) Similarly, the paper finds the condition of $71/84 < \mu < \overline{\mu}$ and $(230 - 84\mu)/159 < \lambda < 1$ in the case of wholesale-and-retail BBP. □

**Proof of Proposition 4.** One obtains the results by comparing the equilibrium outcomes between the case of BBP and the benchmark case.

If $\lambda' < \lambda < 1$ $\left( \lambda' = \frac{-388,936+82,457\mu}{734,993} + \frac{555\sqrt{805,133+791,824\mu-1,049,116\mu^2}}{1,469,986} \right)$, $\Pi_R^{L*} > \Pi_R^{L0*}$

($\Pi_R^{L0*}$ is the profit of the FV retailer in the benchmark case). Other results can also be obtained by comparing profits. □

**Proof of Proposition 5.** In the case of retail BBP,

If

$$\breve{\lambda} < \lambda \leq \overline{\lambda} \ \left( \breve{\lambda} = \frac{683,952}{2,701,207}, \ \overline{\lambda} = \frac{139 + 55\mu}{217} \right),$$

$$\frac{\partial \Pi_{total}^*}{\partial \lambda} = \frac{8s_\Delta(2,701,207\lambda - 227,984(-1 + 2\mu))}{24,950,025} > 0.$$

In the case of wholesale-and-retail BBP,

if

$$\lambda'' < \lambda \leq 1 \left( \lambda'' = \frac{1,599,162}{1,662,835} \right), \ \frac{\partial \Pi_{total}^*}{\partial \lambda} = \frac{s_\Delta(3,325,670\lambda + 1,066,108(1 - 2\mu))}{6,441,444} > 0.$$

□

**Proof of Proposition 6.** In three cases of benchmark case, retail BBP, and wholesale-and-retail BBP, one can easily obtain the results by the first order conditions for the equilibrium profits. Therefore, omit the specific proofs. □

**Proof of Proposition 7.** In the case of retail BBP,

if

$$0 \leq \mu < \hat{\mu} \left( \hat{\mu} = \frac{779 + 262\lambda}{1558} \right), \ \frac{\partial CS^*}{\partial \lambda} = \frac{131s_\Delta(262\lambda + 779(1 - 2\mu))}{24,950,025} > 0.$$

If

$$\widetilde{\lambda} \leq \lambda \leq \overline{\lambda} \ \left( \overline{\lambda} = \frac{139 + 55\mu}{217} \right)$$

or

$$0 < \lambda < \widetilde{\lambda} \left( \widetilde{\lambda} = \frac{5,777,763}{21,643,978} \right), \ 0 < \mu < \widetilde{\mu} \left( \widetilde{\mu} = \frac{1,925,921+21,643,978\lambda}{3,851,842} \right),$$

$$\frac{\partial SW^*}{\partial \lambda} = \frac{s_\Delta(21,643,978\lambda-1,925,921(-1+2\mu))}{24,950,025} > 0.$$

In the case of wholesale-and-retail BBP,

if

$$0 \leq \mu < \hat{\mu} \left( \hat{\mu} = \frac{1}{310}(155 + 229\lambda) \right), \ \frac{\partial CS^*}{\partial \lambda} = \frac{229s_\Delta(155 + 229\lambda - 310\mu)}{3,220,722} > 0.$$

If
$$\widetilde{\lambda}' \leq \lambda \leq 1$$

or

$$0 < \lambda < \widetilde{\lambda}'\left(\widetilde{\lambda}' = \frac{1,705,647}{1,715,276}\right), \ 0 < \mu < \widetilde{\mu}'\left(\widetilde{\mu}' = \frac{568,549+1,715,276\lambda}{1,137,098}\right),$$
$$\frac{\partial SW^*}{\partial \lambda} = \frac{s_\Delta(1,715,276\lambda - 568,549(-1+2\mu))}{3,220,722} > 0.$$

□

**Proof of Lemma 1.** Because two manufacturers are symmetric, the paper only considers one of them. They have identical equilibrium outcomes. Using the same analysis as the main model, $\theta_L = \frac{p_{AN}^H - p_{AO}^L}{s_H - s_L} - \lambda$ in period 2. Following Jing (2016) [23], when a retailer sells both FVs and NEVs, it tends to poach consumers who bought FVs in the first period to buy NEVs in the second period. That is, the consumers in $[0, \theta_L]$ repurchase $L$ in period 2, those in $(\theta_L, \theta_1)$ switch to $H$ in period 2, and those in $[\theta_1, 1]$ repurchase $H$ in period 2. The retailer's profit function in period 2 is as follows.

$$\Pi_{R2}^A = \frac{1}{2}\left[\left(p_{AO}^L - w_{A2}^L\right)\theta_L + \left(p_{AN}^H - w_{A2}^H\right)(\theta_1 - \theta_L) + \left(p_{AO}^H - w_{A2}^H\right)(1 - \theta_1)\right] \quad (A5)$$

We set $p_{AO}^L = V$ and $p_{AO}^H = V + \theta_1 s_H$. That is, the repeat price of Vehicle $L$ fully extracts the utility of consumer 0, and the repeat price of the Vehicle $H$ fully extracts the utility of consumer $\theta_1$. Then, $\partial \Pi_{R2}^A / \partial p_{AN}^H = 0$ and it leads to $p_{AN}^H = \frac{1}{2}[2V + w_{A2}^H - w_{A2}^L + (\theta_1 + \lambda)(s_H - s_L)]$.

The profit function of the manufacturer is as follows.

$$\Pi_{M2}^A\left(w_{A2}^H\right) = \frac{1}{2}\left[\left(w_{A2}^L - c_L\right)\theta_L + \left(w_{A2}^H - c_H\right)(1 - \theta_L)\right]. \quad (A6)$$

(a)  Consider the situation in which the production efficiency of FVs is sufficiently high. In this case, assume the manufacturer charges the lowest wholesale price; i.e., $w_{A2}^L = c_L$. $\partial \Pi_{M2}^A / \partial w_{A2}^H = 0$ leads to $w_{A2}^H = \frac{1}{2}[c_H + c_L + (s_H - s_L)(2 - \theta_1 + \lambda)]$.

In period 1, identify the indifferent consumer $\theta_1$ to the two vehicles.

$$V + \theta_1 s_L - p_{A1}^L + \left[V + \theta_1 s_H + \lambda(s_H - s_L) - p_{AN}^{H*}\right] = V + \theta_1 s_H - p_{A1}^H + \left[V + \theta_1 s_H - p_{AO}^{H*}\right]. \quad (A7)$$

If follows that $\theta_1 = \frac{-c_H + c_L + 4p_{A1}^H - 4p_{A1}^L + (s_H - s_L)(-2+\lambda)}{s_H - 5s_L}$. The profit function of the retailer in period 1 is as follows.

$$\Pi_R^A = \frac{1}{2}\left[\left(p_{A1}^L - w_{A1}^L\right)\theta_1 + \left(p_{A1}^H - w_{A1}^H\right)(1 - \theta_1) + \Pi_{R2}^{A*}\right]. \quad (A8)$$

Set $p_{A1}^L = V$. $\partial \Pi_R^A / \partial p_{A1}^H = 0$ which leads to

$$p_{A1}^H = \frac{\begin{array}{c}(9s_H - 7s_L)(c_H - c_L) + 25s_H^2 - 74s_H s_L + 49s_L^2 + 38V(s_H - s_L) \\ +(4s_H - 20s_L)\left(w_{A1}^H - w_{A1}^L\right) - 5s_H^2\lambda - 8s_H s_L\lambda + 13s_L^2\lambda\end{array}}{38(s_H - s_L)} \quad (A9)$$

The manufacturer's profits are $\Pi_M^A = \frac{1}{2}\left[\left(w_{A1}^L - c_L\right)\theta_L + \left(w_{A1}^H - c_H\right)(1 - \theta_L) + \Pi_{M2}^{A*}\right]$. Also, $w_{A1}^L = c_L$. $\partial \Pi_M^A / \partial w_{A1}^H = 0$ and it leads to $w_{A1}^{H*} = \frac{1}{120}[-23c_H + 11(13c_L + (s_H - s_L)(13 + 5\lambda))]$.

Finally, $\theta_1^* = \frac{1}{15}(17 + 10\lambda - 2\mu)$ and $\theta_L^* = \frac{1}{60}(47 - 5\lambda + 13\mu)$. If $\theta_L^* < \theta_1^*$, $\mu < 1 + \frac{15}{7}\lambda$. Thus, $\theta_1^* > 1$, which is a contradiction.

(b) Now, consider the situation where the production efficiency of NEVs is sufficiently high. In this case, assume the manufacturer charges the lowest wholesale price of the NEV, i.e., $w_{A2}^H = c_H$. $\partial \Pi_{M2}^A / \partial w_{A2}^L = 0$ leads to $w_{A2}^{L*} = \frac{1}{2}[c_H + c_L + (s_H - s_L)(\theta_1 - \lambda)]$.

In period 1,

$$V + \theta_1 s_L - p_{A1}^L + \left[V + \theta_1 s_H + \lambda(s_H - s_L) - p_{AN}^{H*}\right] = V + \theta_1 s_H - p_{A1}^H + \left(V + \theta_1 s_H - p_{AO}^{H*}\right). \tag{A10}$$

If follows that $\theta_1 = \frac{-c_H + c_L + 4p_{A1}^H - 4p_{A1}^L + \lambda(s_H - s_L)}{s_H - 5s_L}$. The profit function of the retailer in period 1 is as follows.

$$\Pi_R^A = \frac{1}{2}\left[\left(p_{A1}^L - w_{A1}^L\right)\theta_1 + \left(p_{A1}^H - w_{A1}^H\right)(1 - \theta_1) + \Pi_{R2}^{A*}\right] \tag{A11}$$

Set $p_{A1}^L = V$. $\partial \Pi_R^A / \partial p_{A1}^H = 0$ which leads to

$$p_{A1}^H = \frac{\begin{array}{c}(9s_H - 7s_L)(c_H - c_L) + 5s_H^2 - 30s_H s_L + 25s_L^2 + 38V(s_H - s_L) \\ +(4s_H - 20s_L)\left(w_{A1}^H - w_{A1}^L\right) - 5s_H^2\lambda - 8s_H s_L\lambda + 13s_L^2\lambda\end{array}}{38(s_H - s_L)} \tag{A12}$$

The manufacturer's profits are $\Pi_M^A = \frac{1}{2}\left[\left(w_{A1}^L - c_L\right)\theta_L + \left(w_{A1}^H - c_H\right)(1 - \theta_L) + \Pi_{M2}^{A*}\right]$. Also, $w_{A1}^H = c_H$. $\partial \Pi_M^A / \partial w_{A1}^L = 0$ leads to $w_{A1}^{L*} = \frac{1}{120}[143c_H - 23c_L - 55(s_H - s_L)(\lambda - 1)]$. Finally, $\theta_1^* = \frac{1}{15}(5 + 10\lambda - 2\mu)$ and $\theta_L^* = \frac{1}{60}(5 - 5\lambda + 13\mu)$.

If $0 < \theta_L^* < \theta_1^* < 1$, $0 < \mu < \frac{1}{7}(5 + 15\lambda)$. This shows that the NEV is sufficiently efficient. □

**Proof of Lemma 2.** This proof is parallel to the basic model. We omit the proof. □

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
