# Peer review of "Competition between New Energy and Fuel Vehicles with Behavior-Based Pricing Strategies When Considering Environmental Concerns and Green Innovation"

_sustainability, doi:10.3390/su16104018_

Round 1

Reviewer 1 Report

Comments and Suggestions for Authors

Reviewer 2 Report

Comments and Suggestions for Authors The article is well written.
The methodology and the empirical part are given in detail.
Maybe a small comment to the conclusion of the article. In the conclusion, it is necessary to give more precise
answers to the research questions.

Reviewer 3 Report

Comments and Suggestions for Authors

The study in this manuscript has certain theoretical and practical significance. They study behavior-based pricing (BBP) strategies between energy vehicles and fuel vehicles in a two-echelon supply chain where consumers are environmentally conscious. The study is interesting and complete. However, the results are too many mathematical equations, abbreviations, tables and figures that are not attractive to the reader and make reading tiring. In addition, some problems need to be modified:

1.       Lines 29, 81 and 88 - It is really necessary to emphasize the words "motivation", "our contribution" and "organization of the paper". In my opinion, this detracts from the flow of the text and should be deleted.

2.       In addition to questions RQ1, R12, RQ3, authors are invited to answer the question "How do consumers' environmental concerns affect the economy? The pillars of sustainability include economic, social and environmental aspects. Therefore, the inclusion of this question fits this journal, which is related to sustainability fields.

3.       Literature review section. The lack of literature is shown in this section, where the contribution of the paper to the scientific community is stated. The last paragraph should be elaborated describing the novelty of the study and real applications.

4.       Figure 4 – The y-axis should be provided by the authors.

5.       Figure 4 – What kind of data is the basis for this figure? This is a result? This is the method? Is not clear to the reader

6.       The text has too many abbreviations, which can damage the integral understanding of the text.

7.       The results and conclusions are interesting, but the presentation is not good enough. Figures and tables should be improved accordingly.

8.       The last paragraph of the conclusion shows the limitations of the paper. Perhaps these limitations should be stated earlier in another part of the text and not in the conclusions.

9.       Discussion can be further elaborated, comparing with other results in literature

10.   Discussion: This section needs to discuss and link the results of the current research with previous studies.

11.   The overall discussion should be further strengthened by including related work recently published in related journals.

12.   The authors are invited to provide the complete data in supplementary material.

13.   Include a discussion of what the results of this study would be in a place where consumers are not environmentally conscious.

Round 2

Reviewer 1 Report

Comments and Suggestions for Authors

The article was corrected correctly. In its present form, it meets the requirements for scientific papers of this type. I thank the authors for precisely following my suggestions. I hope that the authors will continue to conduct research in this area.

Reviewer 3 Report

Comments and Suggestions for Authors

The authors have paid careful attention to the reviewer comments and fully revised their manuscript. The graphs and tables are good supporting information for the text. Overall, I was impressed with the clarity of the revised manuscript.

I have a few minor suggestions, to help the authors polish this work:

1.      Figures are shown in low resolution, please improve it.

2.      Figure 4 - The size of the letters on the y-axis is not large enough to be visible.

3.       By the end of the conclusion, we hope to have a sense of what should happen next. What do you recommend? Is this study feasible in practice? Does it use readily available data, or do the authors have to make a number of assumptions (or use default parameters)? Please give us an idea of how widely this study could be adopted in the community and what the benefits would be.
